# Vanilloid-dependent TRPV1 opening trajectory from cryoEM ensemble analysis

Do Hoon Kwon [1], Feng Zhang[1], Justin G. Fedor [1], Yang Suo[1] & Seok-Yong Lee [1]✉

Single particle cryo-EM often yields multiple protein conformations within a single dataset, but experimentally deducing the temporal relationship of these conformers within a conformational trajectory is not trivial. Here, we use thermal titration methods and cryo-EM in an attempt to obtain temporal resolution of the conformational trajectory of the vanilloid receptor TRPV1 with resiniferatoxin (RTx) bound. Based on our cryo-EM ensemble analysis, RTx binding to TRPV1 appears to induce intracellular gate opening first, followed by selectivity filter dilation, then pore loop rearrangement to reach the final open state. This apparent conformational wave likely arises from the concerted, stepwise, additive structural changes of TRPV1 over many subdomains. Greater understanding of the RTx-mediated long-range allostery of TRPV1 could help further the therapeutic potential of RTx, which is a promising drug candidate for pain relief associated with advanced cancer or knee arthritis.

[1] Department of Biochemistry, Duke University School of Medicine, Durham, NC 27710, USA. ✉email: seok-yong.lee@duke.edu

The structural dynamics of proteins is critical to their function, so visualizing the different conformational states along a reaction coordinate at near atomic-resolution provides rich, unparallel mechanistic information. Advances in single particle cryo-EM have enabled visualization of multiple protein conformations within a single dataset at increasingly higher resolutions[1–3]. Identifying conformations within the same pathway, and resolving their temporal sequence within the cryo-EM ensembles, provides a powerful means to deduce the energetic landscape of macromolecular reaction pathways. However, the exhibited conformational heterogeneity of a sample may be for any number of reasons: different states along the reaction pathway, conformational dynamics (e.g. flexibility), heterogeneity in ligand occupancy, co-existence of on- and off-pathway conformations, and mixtures of folded and partially unfolded proteins. Therefore, experimentally defining the relationship amongst the conformers within the cryo-EM ensembles is challenging despite recent developments in capturing short lived states[4–6].

One fundamental question in ligand-receptor protein interaction is how ligand binding mediates long-range allostery within a receptor protein: through a conformational propagation (or "conformational wave") from the ligand binding site to the effector site, or through concerted global conformational changes via a protein-wide coupled network[7–10]. This question can be better addressed if we dissect the relationship amongst multiple conformations of the ligand bound receptor in the cryo-EM ensembles. To address this question, we chose to study transient receptor vanilloid member 1 (TRPV1), which is the receptor for vanilloids and noxious heat[11]. TRPV1 is an ideal system to study ligand-dependent long range allostery. It is a homotetramer whose transmembrane region is composed of the S1-S4 domains which are peripheral to the central ion-conducting pore domain. Vanilloid binding at the S1-S4 domain allosterically acts upon the pore domain, driving pore opening. Despite the recent deluge of structural information[12–25], the single modality opening of TRPV1 by ligand is not known. The cation-conducting open state was achieved only through the bimodal stimulation [e.g., the ultrapotent vanilloid resiniferatoxin (RTx) and the outer-pore binding double-knot toxin (DkTx)][15,25]. This led to the belief that single modality activation of TRPV1 could not be achieved[25], despite the fact that in electrophysiology studies RTx alone elicits channel open probabilities reaching 1 and allows for conduction of large organic cations[26,27]. Understanding the conformational landscape of RTx-dependent TRPV1 gating is not only fundamentally important, but it also has potential therapeutic impact; RTx is currently being evaluated in phase III clinical trials for pain associated with advanced cancer and arthritis of the knee, with positive preliminary results[28–30]. This raises the promise that RTx could provide a non-opioid solution to intractable chronic inflammatory pain. Here, we employ a thermal titration method to dissect the conformational distributions within and between cryo-EM ensembles in the presence of saturating RTx. Not only do we report the fully open state of full-length TRPV1 via the single modality stimulation of RTx binding, but our analysis also provides insights into the conformational trajectory of RTx-dependent TRPV1 opening, including critical intermediate states, providing a basis for the long range allostery between RTx binding and pore opening.

## Results

**Thermal titration studies of RTx-bound TRPV1 cryo-EM ensembles.** It is known that heat and vanilloid-dependent TRPV1 gating pathways are separate but allosterically linked[13,20]. The heat sensitivity of a process can be quantified by the so-called $Q_{10}$

value which is the ratio between the rates of a process that are separated by 10 °C[31]. For ion diffusion, $Q_{10}$ values are close to 1 (1.1–1.2). For most ion channels a $Q_{10}$ of about 3 is expected, whereas much higher $Q_{10}$ values (>20) are observed for heat-activated thermosensitive TRP channels. The heat sensitivity of TRPV1 is high only above 42 °C ($Q_{10}$ ~20–40) and this noxious heat sensitivity is substantially reduced when ligand gating dominates, i.e. with a saturating concentration of vanilloid[13]. Electrophysiology was used to test the heat sensitivity of RTx-bound TRPV1 (Fig. 1a, b). HEK293T cells expressing rat TRPV1 were dosed with subsaturating concentrations of RTx to elicit varying degrees of currents, followed by a heat ramp from ~10 °C to ~50 °C to provide further stimulation. These currents were then compared relative to currents elicited by a saturating concentration of RTx ($I/I_{50\ nM\ RTx}$), showing a range of stimulation with RTx (~10–100%). Because the apparent RTx potency is subnanomolar ($EC_{50}$ ~30 pM)[32] and its stickiness precludes precise control at low concentrations, this ratio of $I/I_{50\ nM\ RTx}$ can be viewed as a proxy for RTx occupancy. Based on the $Q_{10}$ plot in Fig. 1c, RTx-bound TRPV1 (>20% RTx occupancy) loses sensitivity to heat (blue trace), exhibiting a $Q_{10}$ similar to the level of non-heat sensor ion channels ($Q_{10}$~3), while the heat sensitivity of RTx-bound TRPV1 at lower temperatures (red trace) is similar to that of apo TRPV1 ($Q_{10}$~1.6, Fig. 1c). This is ideal for studying TRPV1 ligand-gating because under the saturating RTx conditions when RTx occupancy is maximized, thermal energy can be used to more gradually shift the conformation of TRPV1 toward the final open state without invoking the high noxious heat sensitivity. This may be because RTx binding ablates high noxious heat sensitivity or RTx-gating of TRPV1 dominates under this condition. In this condition we do not need to invoke partial occupancy and high heat sensitivity for our structural studies.

Full-length rat TRPV1 was purified and reconstituted into nanodiscs at 4 °C as described[19]. RTx was incubated with nanodisc-reconstituted TRPV1 for 30 min at 4 °C prior to grid freezing. Cryo-EM 3D reconstructions proceeded from manually picked particles which were used to generate templates for template-based particle picking (Supplementary Fig. 1). Various strategies of 3D classification of the 4 °C dataset consistently resulted in three distinct classes with nearly equal particle distributions (Supplementary Fig. 2) to overall excellent quality (Supplementary Fig. 3). Each 3D class at 4 °C exhibits differing extents of pore opening, indicating that they may represent different conformational states along the ligand-dependent activation pathway. 3D classification was conducted using C1 symmetry in order to address any asymmetric occupancy of RTx, however we found no evidence for this and observed equal occupancy at all four sites. Since a saturating amount of RTx was used, and all classes were occupied with RTx, an RTx titration would not help probe the relationship between the classes. Therefore, we decided to use thermal energy to redistribute class populations. We collected two additional datasets for which RTx-TRPV1 was incubated at 25 °C and 48 °C for 30 s before flash freezing. Similar processing strategies were applied to cryo-EM 3D reconstructions of the RTx-TRPV1 complex at 25 °C and at 48 °C (Supplementary Fig. 1). We found that 3D classification of the dataset at 25 °C converged upon two classes and the dataset at 48 °C resulted in one class. For the three classes of RTx-TRPV1 at 4 °C, we assigned class I, class II, and class III according to the size of the pore opening (class I being the narrowest). Notably, the two classes at 25 °C are very similar to classes II and III at 4 °C and the single class at 48 °C is very similar to the class III at 4 °C (Figs. 1 and 2, Supplementary Fig. 4). Correlating the class distributions with increasing temperature, we propose that the three conformational states in the cryo-EM ensembles of RTx-TRPV1 at 4 °C represent distinct conformations along the RTx-

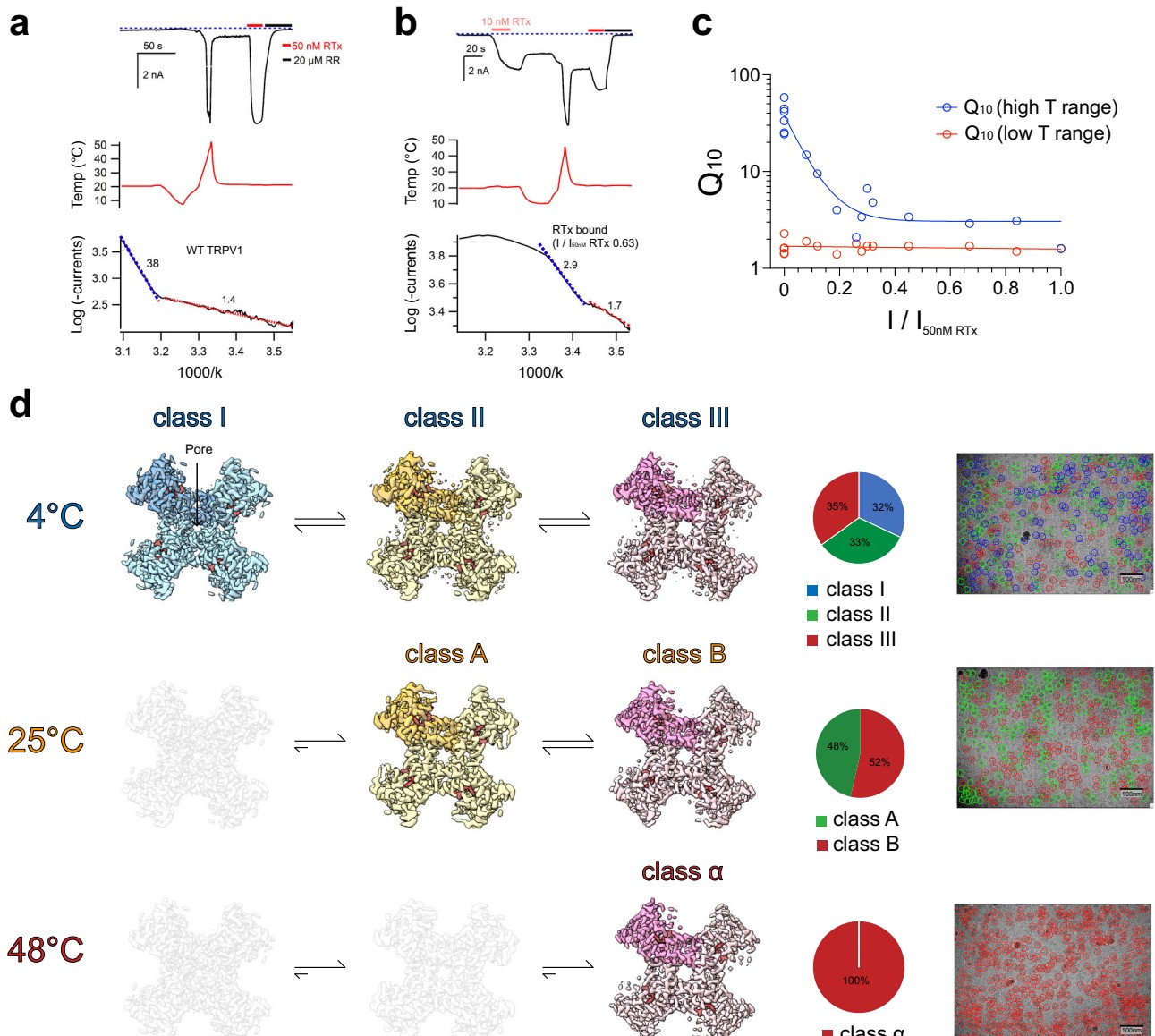

**Fig. 1 Thermal titration cryo-EM experiment and the effect of RTx on TRPV1 heat sensitivity. a** A representative macroscopic current time-course (top panel) recorded from a HEK-293T cell expressing rat TRPV1 in response to the temperature ramp (10–50 °C) at a membrane potential of −60 mV and then followed by a saturating concentration of RTx (50 nM) and 20 μM ruthenium red (RR). The dashed line indicates zero current. The recorded temperature is shown in the middle panel. The Arrhenius plot for the temperature activation was shown in the bottom panel. Fitted $Q_{10}$ values for high (blue line) and low (red line) temperature ranges are shown. **b** A representative time-course recording for RTx-bound TRPV1 temperature sensitivity. First the channel was challenged by 10 nM RTx for ~20 s followed by a temperature ramp (10–48 °C), then a saturating concentration of RTx (50 nM) was introduced, and finally RR (20 μM) was applied to completely block the channel. The dashed line indicates zero current. The recorded temperature is shown in the middle panel and the Arrhenius plot for the temperature activation is shown in the bottom panel. Fitted $Q_{10}$ values for high and low temperature (T) ranges are shown. **c** $Q_{10}$ values as a function of $I/I_{50nM\ RTx}$ for low and high temperature ranges. Each experiment was conducted as shown in **a** and **b**. The low T range $Q_{10}$ value is steady at 1.7, while the high T range $Q_{10}$ rapidly collapses from ~38 to ~3. Each pair of high and low temperature sensitivity data points represents independent time-course recordings from individual cells ($n = 17$ cells). Source data are provided as a Source Data file. **d** Representative micrographs of TRPV1 recorded in the presence of 50 μM RTx at 4 °C, 25 °C and 48 °C, respectively. Cryo-EM maps of RTx-TRPV1 determined at 4 °C (class I, class II, and class III), 25 °C (class A and class B), and 48 °C (class α). Note the differences between central pore sizes amongst different classes at 4 °C. The classes not found in each dataset are shown as transparent. The pie charts depict particle distributions among classes for each dataset along with representative micrographs. Each pie chart represents an average value for four independent data processes (Supplementary Fig. 2a, b).

dependent gating pathway and that their temporal sequence proceeds from class I through class II to class III, following their expectedly increasing energetic states (Fig. 1d). We then applied a basic thermodynamic analysis to the populations of particles in each state over each temperature, calculating relative equilibrium constants where possible, as well as enthalpies for temperature-

dependent shifts in equilibria (Supplementary Fig. 5). We tentatively assume that class I, II, and III are closed, intermediate, and open states. Our electrophysiology experiments indicate that RTx binding renders TRPV1 relatively insensitive to heat, with $Q_{10}$ (high T range) rapidly collapsing from ~38 (no RTx) to ~3, and $Q_{10}$ (low T range) not deviating from 1.7 (Fig. 1c). This bore

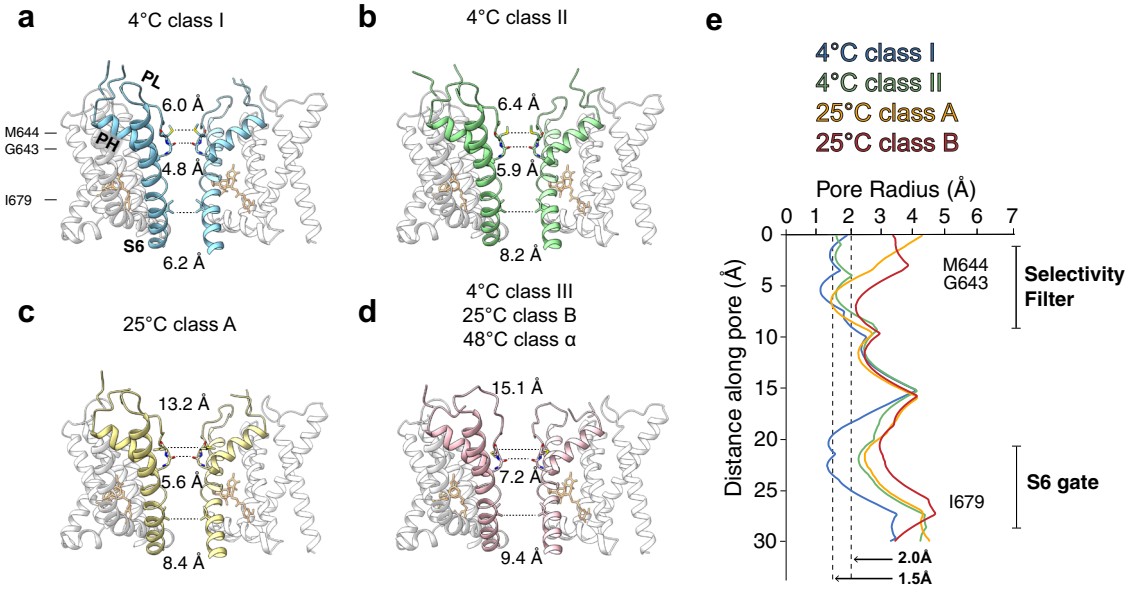

**Fig. 2 Permeation pathway analysis of the pore domains of TRPV1[4 °C, class I], TRPV1[4 °C, class II], TRPV1[25 °C, class A], TRPV1[4 °C, class III], TRPV1[25 °C, class B], and TRPV1[48 °C, class α]. a–d** Only two subunits are shown for clarity, with S6 helix (S6), pore loop (PL) and pore helix (PH) as indicated. Diagonal distances at the two narrowest restriction points are shown. **e** Pore radii calculated using the HOLE program for representative TRPV1 structures as color-coded. The radius of water is 1.5 Å, while 2.0 Å is considered the minimal radius for a hydrophobic gate to be open. Residues corresponding to the SF (M644 and G643) and the S6 gate (I679) are denoted.

out in the structural thermodynamic analysis, where no large shifts in equilibria were observed with temperature. Upon heating, only modest shifts in equilibria push the complex to a progressively more open state.

**Conformational pathway of TRPV1 gating by RTx.** All six 3D reconstructions were refined to overall excellent quality (3.04–3.45 Å) for which atomic models were built (Supplementary Fig. 3 and Supplementary Table 1). TRPV1 is a homotetramer, where each protomer has an N-terminal cytosolic ankyrin repeat domain (ARD) and a transmembrane domain (TMD) of six transmembrane helices (S1-S6). The TMD is composed of a voltage sensor-like S1-S4 domain (VSLD), a pore domain (S5, the turret, the pore helix (PH), the selectivity filter (SF), the pore loop (PL), and S6), and the amphipathic TRP helix. The ARD and TMD are linked via the coupling domain (CD), which includes a helix-turn-helix motif (HTH_CD), a β-sheet (β_CD), the pre-S1helix (pre-S1_CD), and a C-terminal domain (CTD) (Supplementary Fig. 6a).

We performed HOLE analyses of the six models to calculate the sizes of the S6 gate (I679) and the SF (G643 and M644) constriction points. A gate composed of hydrophobic residues must have a pore radius of at least 2.0 Å (the diagonal distance ~7.6 Å) to be able to conduct partially hydrated ions, while a gate comprised of hydrophilic residues may be narrower[33]. Based on this criterion, the 4 °C class I represents a nonconducting, closed state whereas class III represents a conducting, open state (Fig. 2). The models of the 25 °C class B reconstruction and the 48 °C reconstruction are nearly indistinguishable to that of the 4 °C class III reconstruction (Cα r.m.s.d. <0.45 Å), suggesting that they all represent the open state (Fig. 2 and Supplementary Fig. 4). Notably, the models from the 4 °C class II and 25 °C class A reconstructions have similar overall conformations (Cα r.m.s.d. ~0.6 Å), however local differences near the SF and the CD suggests that they represent two intermediate substates (see the

analysis below) (Supplementary Fig. 4a, b). We assigned (i) the class I at 4 °C as a closed state (TRPV1[C,RTx]), (ii) the class II at 4 °C as an intermediate, closed state (TRPV1[IC,RTx]), (iii) the class III at 4 °C as the final open state (TRPV1[O,RTx]), (iv) the class A at 25 °C as an intermediate, open state (TRPV1[IO,RTx]), (v) the class B at 25 °C as the final open state (TRPV1[O,RTx]), and (vi) the reconstruction at 48 °C as the final open state (TRPV1[O,RTx]) (Fig. 3a).

Having assigned the order of observed transitions within the cryo-EM ensembles, we analyzed the sequential conformational changes of RTx-dependent TRPV1 gating. Comparing the RTx-bound closed state of TRPV1 (TRPV1[C,RTx]) with the published apo TRPV1 structure shows that RTx binding induces slight local changes near the vanilloid binding site of the S1-S4 domain (S3, S4b, the S4-S5 linker), but otherwise the conformation remains similar to the apo TRPV1 structure (Supplementary Fig. 6). Compared to TRPV1[C,RTx], the 4 °C intermediate state (TRPV1[IC,RTx]) exhibits rotation of the TRP domain while the C-terminal half of S6 (S6b) rotates and tilts away from the ion permeation pathway so that the diagonal distance between the side chains of I679 in the S6 gate is larger than 8 Å, consistent with an open conformation for the S6 gate (Figs. 2, 3b, and 4). Rotation of S6b occurs at the π helical region between S6a and S6b (Fig. 3a)[34]. However, the SF of the 4 °C intermediate state adopts a nonconducting conformation, as there is minimal change at the SF with the diagonal distance between the side chains of M644 of 6.4 Å. Because the backbone carbonyl groups of G643 in the SF can coordinate dehydrated Na+, as evidenced by the cryo-EM density maps of the TRPV1[C,RTx] and the published apo TRPV1 (Supplementary Fig. 7)[19], the only residue that can potentially block ion permeation at the SF is M644. Intriguingly, in the 25 °C intermediate state, M644 is flipped away from the central ion permeation pathway such that the inter-M644 distance is ~13 Å, concomitant with a slight SF displacement (Figs. 3c and 4). Therefore, because both the SF and the S6 gate adopt conformations consistent with conductive states and

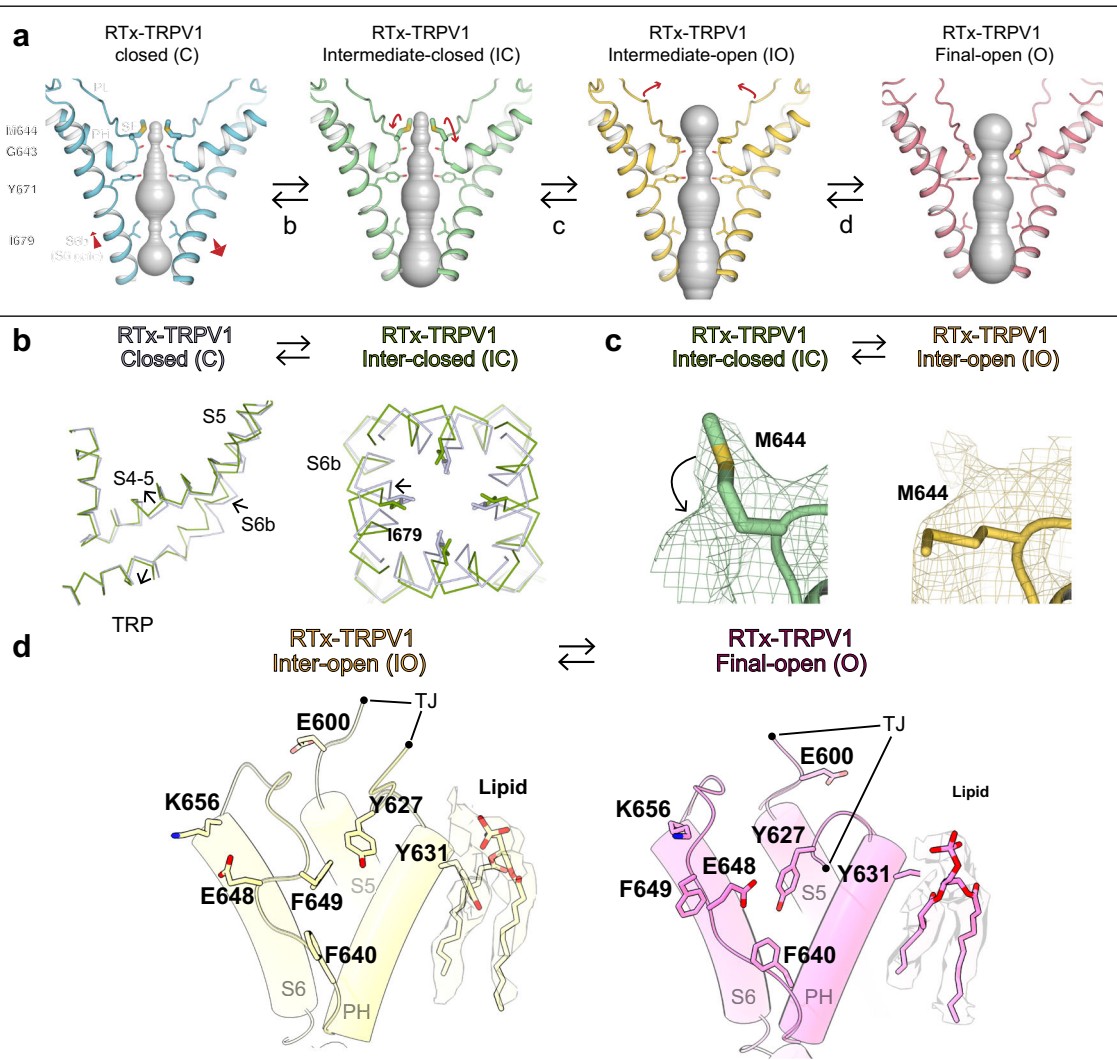

**Fig. 3 RTx-dependent conformational trajectory of TRPV1. a** Comparison of the pore domain structures, only two subunits are shown for clarity, with the S6 gate (S6b), selectivity filter (SF), pore loop (PL) and pore helix (PH) as indicated. The pore profiles are shown as surfaces (gray). The red arrows indicate direction of movement. **b** Comparison of TRPV1$^{C,RTx}$ (gray) and TRPV1$^{IC,RTx}$ (green) structures (left) and close-up view of TRPV1$^{C,RTx}$ and TRPV1$^{IC,RTx}$ pore region (right). **c** The cryo-EM densities and the models for M644 in TRPV1$^{IC,RTx}$ (green) and TRPV1$^{IO,RTx}$ (gold). The cryo-EM map thresholdings are 0.03, and 0.04, respectively. **d** Comparison of TRPV1$^{IO, RTx}$ (gold) and TRPV1$^{O, RTx}$ (pink) outer pore region. Representative residues showing large motions are shown as sticks. TJ, turret junction. Phospholipids are shown as sticks and cryo-EM densities, with thresholding at 0.035 and 0.029, respectively.

the M644 flipped conformation is observed in the final open state (TRPV1$^{O,RTx}$, Fig. 4), we tentatively assign the model from the 25 °C class A reconstruction as an intermediate open state (TRPV1$^{IO,RTx}$), while assigning the corresponding 4 °C class II reconstruction as an intermediate closed state (TRPV1$^{IC,RTx}$). We reason that the two intermediate states at 4 °C and 25 °C represent substates in which their energetic and conformational differences are too small to be separated by cryo-EM 3D classification. Upon lowering the contour level of the cryo-EM density maps, we observe density for both M644 rotamers in both intermediate state 3D reconstructions, but the density corresponding to the non-conductive conformation predominates at 4 °C whereas the conductive conformation predominates at 25 °C, supporting our hypothesis (Fig. 3c). Similar to the M644 flipping, slight movement of the CD toward the pore domain was observed in the 25 °C intermediate state (Fig. 5d and Supplementary Fig. 4a).

In the final open state (TRPV1$^{O,RTx}$) the SF and S6 gate are further dilated compared to the intermediate open state

(TRPV1$^{IO,RTx}$): the diagonal distance at I679 on S6 is increased from 8.4 Å to 9.4 Å and those at M644 and G643 are increased from ~13 Å to ~15 Å and 5.6 Å to 7.2 Å, respectively (Fig. 2). The conformational transition from TRPV1$^{IO,RTx}$ to TRPV1$^{O,RTx}$ is primarily rearrangement of the outer pore - including the PL, PH, and the junctional regions between S5, turret, and S6 (the turret junction: TJ) (Fig. 3d). Displacement of the PL from the PH creates a large gap between them, which is occupied by the TJ. Many sidechains of the PL, PH, and TJ change their rotameric conformations to accommodate these rearrangements, including F640 (PH), E648 (PL), F649 (PL), K656 (PL), E600 (TJ) and Y627 (TJ). Many of these conformationally dynamic residues in the outer pore are demonstrably important for TRPV1 vanilloid gating[14,35–37]. We also observed a concerted conformational change of Y631 (PH) and a phospholipid bound at the interface between two subunits of the outer pore (Fig. 3d). It appears that rearrangements of the outer pore are needed to stabilize the open pore conformation. In the cytoplasmic side, we also observed noticeable conformational differences at the ARD and ARD/CTD

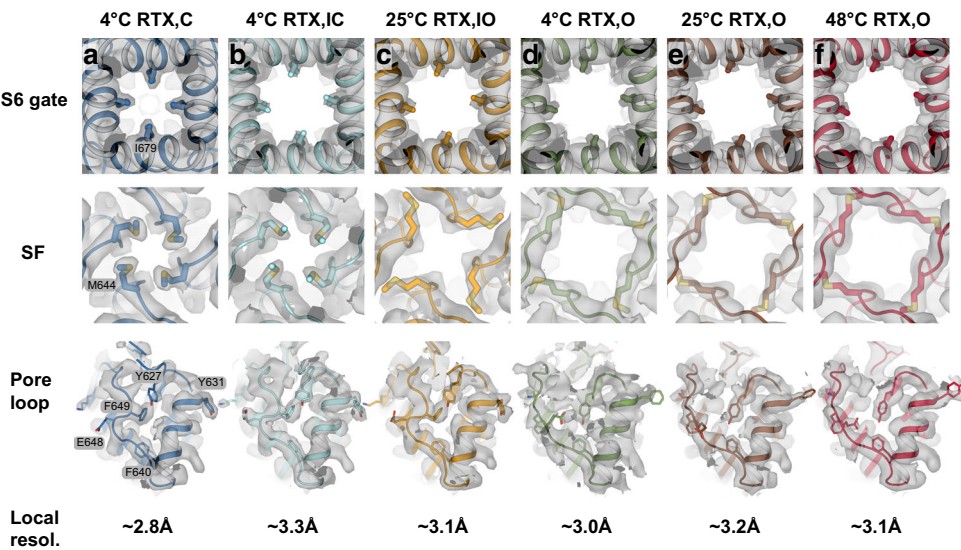

**Fig. 4 Pore comparison across TRPV1^C, RTx, TRPV1^IC, RTx, TRPV1^IO, RTx, and TRPV1^O, RTx.** The cryo-EM densities (grey surface) and respective models (cartoon) depicting bottom-up views of the S6 gate (top), top-down views of the selectivity filter (middle), top-down views of the monomeric outer pore (bottom), and local estimated resolutions for TRPV1^C, RTx **a**, blue, thresholding 0.12); TRPV1^IC, RTx **b**, cyan, thresholding 0.035); TRPV1^IO, RTx **c**, orange, thresholding 0.09); TRPV1^O, RTx,4 °C **d**, green, thresholding 0.1); TRPV1^O, RTx,25 °C **e**, brown, thresholding 0.08); and TRPV1^O, RTx,48 °C **f**, red, thresholding 0.033).

interface in the TRPV1^O,RTx. Taken together, we propose that the conformational steps for RTx-dependent TRPV1 pore opening proceeds first through S6 gate opening, SF opening (via M644), then rearrangement of the PL and the outer pore (Fig. 3). The snapshots of these regions in the cryo-EM maps illustrate a plausible conformational trajectory of RTx-dependent TRPV1 gating (Fig. 4).

**Long-range allostery by RTx of TRPV1.** The visualized RTx-dependent TRPV1 conformational trajectory (Figs. 3 and 4) provides a unique opportunity to address the long-standing question of how ligand binding to a protein achieves long-range allostery. Do the conformation changes initiated upon RTx binding propagate from the ligand binding site to the pore like a conformational wave, or in a concerted fashion[9,18,38,39] ? Using the hydroxyl group of Y511 of the RTx binding site as the reference point, distances to key sites along the pore are ~20 Å (the S6 gate, I679), ~28 Å (SF, M644), ~40 Å (PL, E651), ~48 Å (TJ, E600) and to those within the cytoplasmic domain are ~25 Å (CD, M412), ~45 Å (ARD/CTD, C257) (Fig. 5a). Intriguingly, the order of significant conformational changes in the pore and the cytoplasmic domain correlates well with the distances to the RTx binding site, as if the conformational changes radiate from the RTx binding site, consistent with the conformational wave idea proposed by previous studies[18].

However, overlaying all the conformations of each subdomain, including the RTx binding site, SF, S6 gate, and ARD/CD (Fig. 5b–e), suggests that individual subdomain elements move together in a concerted manner, with large long-range conformational changes appearing to occur as sequential stepwise motions in accordance with the distance from the ligand-binding site. For example, although the SF dilation appears to take place in the last step in the conformational trajectory, progressive movements and dilations of the SF take place in each step (Fig. 5c). Similar observations were made for the ARD/CD as well as the S6 gate. Taken together, this observation strongly suggests that all these subdomains (the SF, S6 gate, PL, PH, TJ, and ARD/CD) are tightly coupled together, and conformational transitions occur as collective motions of individual subdomains in each discrete step.

This requires an interaction network that couples multiple subdomains throughout the channel.

Amongst the multiple coupled subdomains of the interaction network, we pay particular attention to the coupling between the outer pore and the S6 gate, as motions of S6 (the S6 gate) is apparently coupled to tilting of the PH and dilation of the outer pore. It is possible that the S1-S4 domain plays a role in the coupling, but we do not observe obvious conformational changes in the S1-S4 domain over the RTx-dependent conformational trajectory of TRPV1 (Supplementary Fig. 8). Instead, we realized that there is a tripartite interaction amongst the PH (T641), S5 (Y584), and S6 (Y666), which may play a role in coupling between the outer pore and the S6 gate (Fig. 6). In the closed state T641 (PH) and Y584 (S5) interact with each other, but they do not interact with Y666 (S6) which is concomitant with the first S6 dilation. This tripartite interaction tightens, with each of the three residues interacting with each other in the intermediate and final open states, indicating the triad's importance for SF dilation and outer pore rearrangement in the final open state (Fig. 6). We have previously shown that TRPV1 T641A preserves the Na⁺ conduction, but the large cation YO-PRO-1 (MW of 376 Da) significantly reduces Na⁺ currents[27]. This observation is consistent with the idea that T641A affects the final open state and thus limits the SF dilation. We found that, in the presence of RTx, introducing either the Y584F mutation on S5 or the T641A mutation on PH renders Na⁺ currents highly sensitive to the presence of the large cation YO-PRO-1. This suggests the inability of these mutants to fully open such that YO-PRO-1 acts as a blocker (Fig. 6b–e). Y666A[21] or Y666F (in this study) do not generate currents to a detectable level. Together, the results from our previous and current functional studies are consistent with the role of the tripartite interaction network in the coupling between the outer pore and the S6 gate in TRPV1 gating. Similar mutational effects were observed in TRPV2 when two of the corresponding tripartite interactions were disrupted, suggesting the conservation of the coupling interaction network between TRPV1 and TRPV2[27].

**Distinct conformational paths for stimulus-specific TRPV1 gating.** RTx is an ultrapotent TRPV1 agonist with an apparent affinity 3000 fold tighter than capsaicin[32]. RTx irreversibly opens

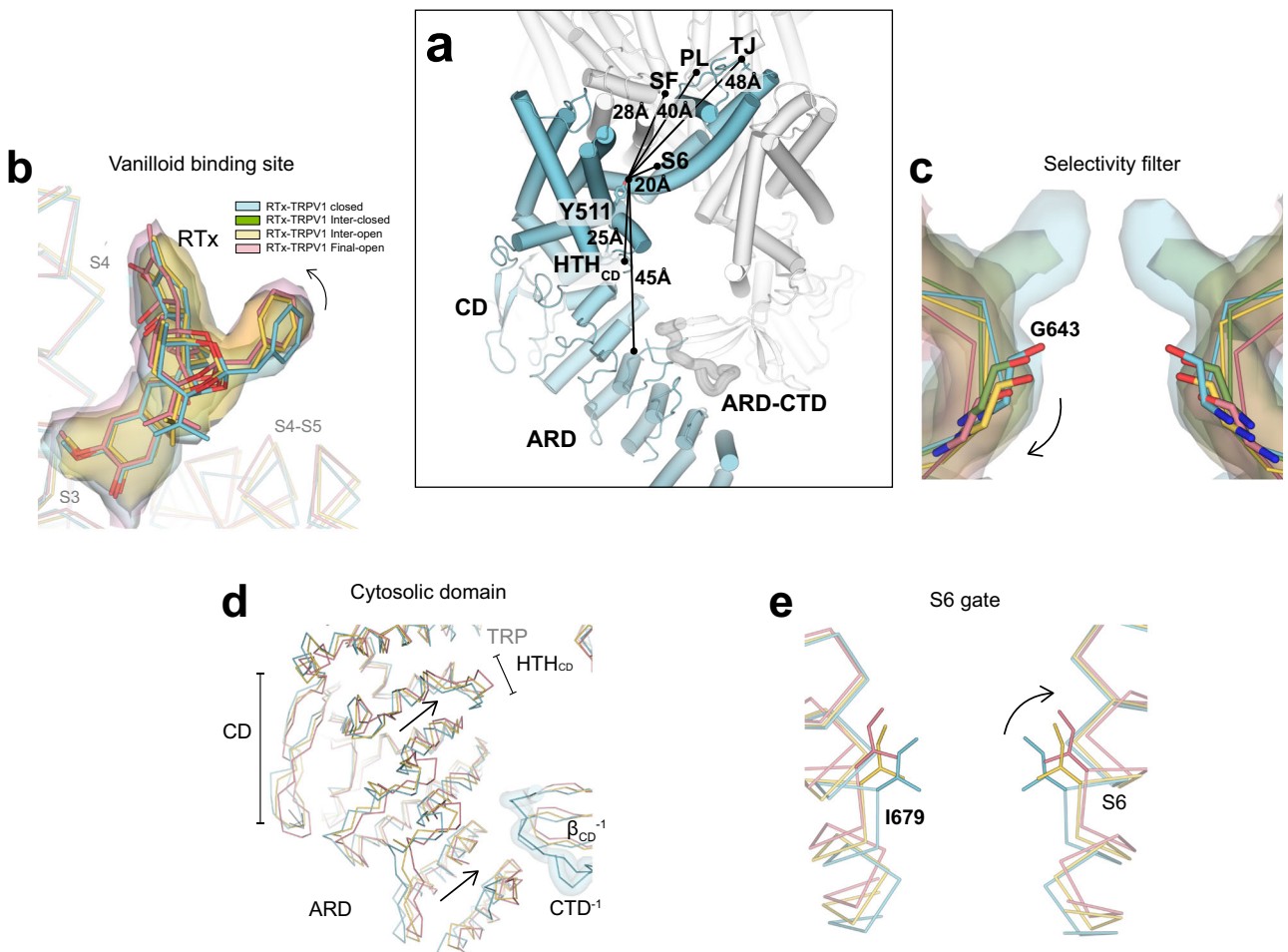

**Fig. 5 RTx-dependent long-range conformational changes in TRPV1. a** Cylinder representation of TRPV1 in turquoise (one subunit) and gray (the rest of the channel). The approximate distances from the RTx binding site (reference residue Y511) to subdomains are shown. **b** The cryo-EM densities (surface) and respective models (sticks) depicting close-up views of the vanilloid binding sites in TRPV1$^{C, RTx}$ (skyblue), thresholding 0.19, TRPV1$^{IO, RTx}$ (yellow), thresholding 0.04, and TRPV1$^{O, RTx}$ (pink), thresholding 0.033. **c** The cryo-EM densities (surface) and respective models (sticks) depicting close-up views of the selectivity filter in TRPV1$^{C,RTx}$ (skyblue), thresholding 0.19, TRPV1$^{IC,RTx}$ (green), thresholding 0.04, TRPV1$^{IO,RTx}$ (yellow), thresholding 0.1, and TRPV1$^{O,RTx}$ (pink), thresholding 0.033. **d–e** Close-up view of the overlays of TRPV1$^{C, RTx}$ (skyblue), TRPV1$^{IO, RTx}$ (yellow), and TRPV1$^{O, RTx}$ (pink) regarding the cytoplasmic domain, and S6 gate **e**, respectively.

TRPV1 without desensitization[26], which is distinct from capsaicin. We compared the RTx conformational trajectory with that of heat-activated TRPV1, sensitized by capsaicin. It was previously shown that capsaicin-bound TRPV1 in nanodiscs can only be opened under noxious heat (48 °C)[19], whereas a substantial proportion of TRPV1 is open even at 4 °C for RTx-bound TRPV1 in nanodiscs. This structural observation is consistent with electrophysiological experiments which show that capsaicin-sensitized, closed TRPV1 retains heat sensitivity ($Q_{10} > 15$ in oocytes)[19] while RTx-bound TRPV1 has substantially reduced heat sensitivity (in this study). Ultrapotent and irreversible binding of RTx binding to TRPV1 likely contributes to the reduced heat sensitivity of TRPV1. Structural analyses show that RTx-activation and heat activation occur through distinct conformational pathways. The salt bridge between S4 (R557) and S4-S5 linker (E570) was proposed to be key in gate opening[16]. For heat-activation of capsaicin-sensitized TRPV1, there is an intermediate state where the S6 gate adopts a non-conducting conformation while the S4/S4-S5 salt bridge is formed. However, RTx-dependent conformational transitions do not involve this intermediate, instead proceeding through a state where the S6 gate dilates first before salt bridge formation (Supplementary Fig. 9).

We compared our structures with the recent structure of the full-length TRPV1 in complex with RTx that was proposed to be in a partially open state ($O_1$). HOLE analysis of the S6 gate opening and the SF (diagonal distances 6.8 Å for I679 and 4.9 Å for M644) suggests that this structure adopts a non-conducting state (Supplementary Fig. 10). Exhibiting a narrower S6 and SF opening than the intermediate-closed state (diagonal distances 8.2 Å and 6.4 Å, respectively) places this structure between the closed and the intermediate-closed states in our conformational trajectory (Fig. 2).

Comparison of our RTx-bound final open TRPV1 structure with both truncated and full-length DkTx/RTx-activated TRPV1 structures reveals that the RTx-induced outer pore structure is distinct from those of DkTx/RTx. The PL and TJ in the RTx-open TRPV1 structure adopt distinct conformations from those structures with DkTx and RTx (Supplementary Fig. 11). Consistent with this observation, it was previously shown that activation by RTx or DkTx elicit different single channel conductances from TRPV1[40]. The observed conformational difference is likely due to the fact that DkTx binding to the PL and TJ and the displacement of a phospholipid at the subunit-subunit interface of the outer pore of TRPV1. This phospholipid

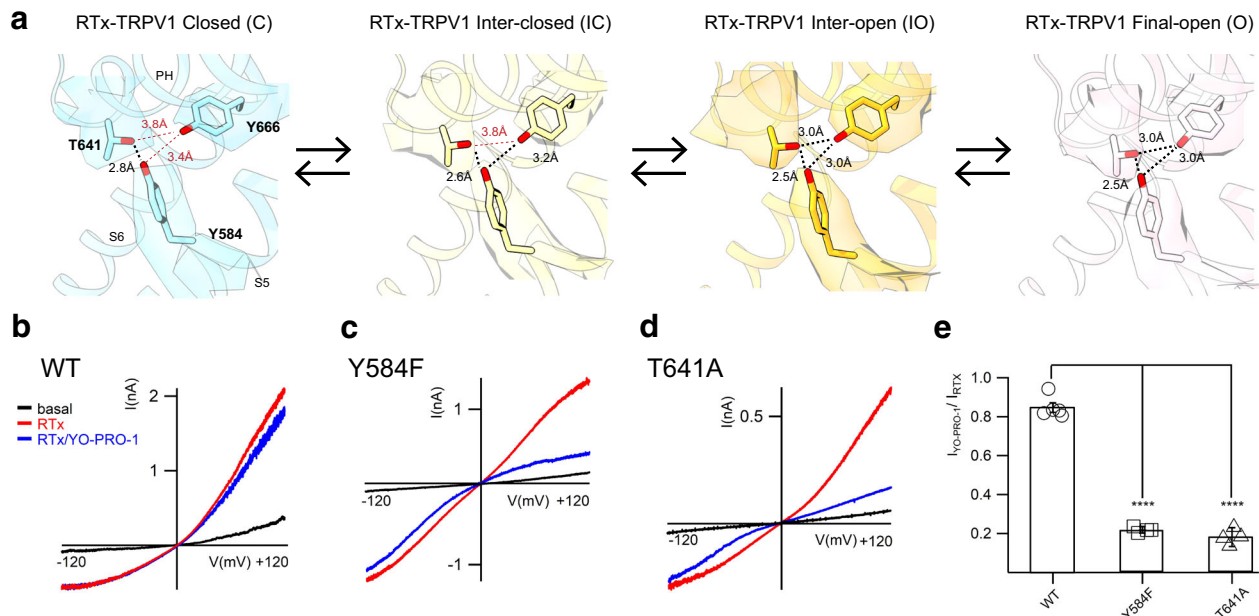

**Fig. 6 PH-S5-S6 triad hydrogen bond network in TRPV1 for RTx gating. a** The cryo-EM maps (surface) and respective models (sticks) depicting the tripartite hydrogen bond network of PH-S5-S6 in TRPV1$^{C, RTx}$ (skyblue), thresholding 0.15, TRPV1$^{IC, RTx}$ (yellow), thresholding 0.045, TRPV1$^{IO, RTx}$ (gold), thresholding 0.1, and TRPV1$^{O, RTx}$ (pink), thresholding 0.04. The black dotted-lines indicate hydrogen bonds. The red dotted-lines indicate distance measurements between atoms where hydrogen bonds are broken. **b–e** TRPV1 Y584F and T641A reduce large cation permeabilty (YO-PRO-1, M.W. 376 Da) in the presence of RTx. Representative inside-out current traces of TRPV1 WT **b**, TRPV1 Y584F **c**, and TRPV1 T641A **d**. Current traces for basal, RTx (200 nM) activation (red trace) and intracellular application of 10 μM YO-PRO-1 (blue trace). **e** Summary of current inhibition by YO-PRO-1 (10 μM) of TRPV1 WT, TRPV1 Y584F and TRPV1 T641A after application of a saturating concentration of RTx (200 nM). Data are presented as mean ± s.e.m.; $P < 0.0001$ for T641A and Y584F, two-tailed unpaired Student's t test. (For WT, $n = 5$, for Y584F and T641A, $n = 4$). Source data are provided as a Source Data file.

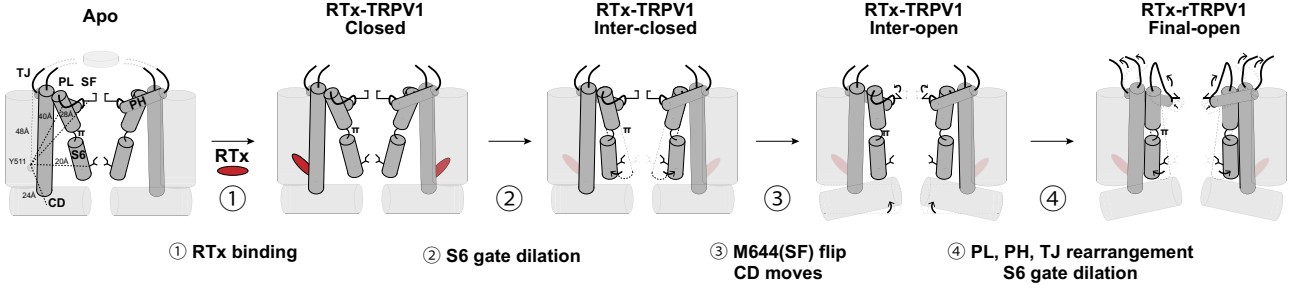

**Fig. 7 RTx-mediated TRPV1 gating mechanism.** In the unstimulated apo state, the channel is closed both at the selectivity filter and S6 gate. **1** Initially, RTx binds with no significant conformational changes. **2** RTx binding induces S6 gate dilation. **3** The M644 sidechain flips outward, and the CD moves towards the channel core. **4** Finally, rearrangement of the PL, PH, TJ results in further dilation of the S6 gate.

binds and rearranges during the final opening step of RTx-bound TRPV1 (Fig. 3d). Notably, in addition to the structural difference of the outer pore, the triad interactions among T641 (PH), Y584 (S5), and Y666 (S6) are broken in the DkTx and RTx bound TRPV1 structures (Supplementary Fig. 11e, f)[16]. Since DkTx and RTx bind to and affect the outer pore and the S6 gate respectively, the coupling between the outer pore and S6 gate is broken or altered.

Taken together, our structural studies show that the RTx-dependent TRPV1 gating pathway is distinct from those dependent upon DkTx or heat.

## Discussion

We report the fully conducting structure for TRPV1 through single modality stimulation by the vanilloid, RTx. Using a method of thermal titration and cryo-EM, we constructed the energetic landscape for vanilloid-dependent TRPV1 gating (Fig. 7).

Tracking the population shifts from classes I, II, and III at 4 °C to classes A and B at 25 °C and finally class α at 48 °C suggested the TRPV1 opening trajectory (class I to class II/A to class III/B/α) without relying on pore diameter. Comparison with class III regarding the M644 sidechain conformation facilitated the sequential placement of the two intermediate sub-states (inter-open and inter-closed) (Figs. 3a and 4). TRPV1 undergoes a wave-like conformational propagation centered around the RTx binding site, first through S6 gate opening, SF opening, then rearrangement of the PL and the outer pore (with further S6 gate and SF opening), which contrasts with the two-state transition observed for the Na$^+$-dependent K$^+$ channel[41]. We found that this apparent conformational wave is due to additive, concerted conformational changes across many subdomains, providing insights into the mechanism of ligand-mediated long-range conformational changes. Last, we found that the RTx-dependent conformational landscape is distinct from that of DkTx or heat

activation. Our studies show that TRPV1 relies upon an extensively coupled interaction network in which the nature of energetic coupling is highly sensitive to different stimuli. It is possible that RTx binding ablates the high noxious temperature sensitivity of TRPV1 or that RTx-dependent gating dominates TRPV1 gating. Also, it is possible there may be effects from partial RTx occupancy, but we see no evidence for this in our data. Our results may not fully reflect the physiological condition due to unavoidable differences between the in vitro and in vivo environments. However, despite the potential difference between our cryo-EM and cell-based electrophysiological experiments, we would like to emphasize that our findings help fill the gap to prove that a single modality is sufficient to fully activate TRPV1 both in vivo and in vitro, as opposed to previous studies[15,16,25].

We believe that our cryo-EM method for probing the temporal sequence of conformations within cryo-EM ensembles is an approach applicable to many other systems. Previously, thermal titration was used to reach alternative states[42], but our method aims to shift existing populations within a single experimental setup to reconstruct the conformational continuum of a given system. Since the ligand-dependent conformational landscape of macromolecules is thermally distributed, with heat sensitivities comparable to the system studied here ($Q_{10}$ ~1.5–3), our thermal titration method can, in principle, be used to probe the conformational pathway of many other proteins and macromolecules. Similar cryo-EM ensemble analysis and thermodynamic modelling would also be applicable to such systems, and ever larger datasets could reveal rare highly energetic states. We recognize that the limitation of our study lies in the difficulty to capture rare states. Recent developments in capturing short-lived states[4] and further improved processing methods[43] will help increase the temporal range and sensitivity to probe conformational landscapes within cryo-EM ensembles.

## Methods

**TRPV1 protein expression and purification**. The full-length wild-type *Rattus norvegicus* (rat) TRPV1 protein was expressed, purified and reconstituted into nanodiscs as previously reported, with minor modifications[19]. In short, the wild-type construct was cloned into pEG-BacMam[44] and transformed into DH10Bac *E. coli* cells for bacmid generation. Baculovirus was produced following the manufacturer's protocol (Invitrogen). For TRPV1 protein expression, 6% (v/v) P3 baculovirus was used to infect HEK293S GnTI- cells maintained in Freestyle 293 media (Gibco) supplemented with 2% (v/v) FBS in 8% $CO_2$ at 37 °C. After 16 h post-infection, the culture was dosed with 10 mM sodium butyrate and the temperature was reduced to 30 °C to boost protein expression. The cells were harvested by centrifugation at 550 × g after 66–72 h post-infection and were resuspended in lysis buffer (20 mM Tris pH 8, 150 mM NaCl, 1% (w/v) digitonin, 12 μg mL$^{-1}$ leupeptin, 12 μg mL$^{-1}$ pepstatin, 12 μg mL$^{-1}$ aprotinin, 4 μg mL$^{-1}$ DNase I, 2.0 mM phenylmethylsulphonyl fluoride (PMSF), 2 mM dithiothreitol (DTT)). The protein was solubilized at 4 °C for 1.5 h, followed by centrifugation at 16,000 × g for 30 min to remove the insoluble material. Batch binding of the supernatant to Anti-FLAG M2 resin (Sigma-Aldrich) was performed for 1 h at 4 °C under gentle agitation. Subsequently, the resin was packed on a gravity-flow column (Bio-Rad) and washed with ten column volumes of buffer A (20 mM Tris pH 8, 150 mM NaCl, 0.06 % digitonin, and 2 mM DTT) and eluted with five column volumes of the buffer A supplemented with 0.12 mg mL$^{-1}$ FLAG peptide (GenScript). The elution was collected, concentrated, and subjected to nanodisc reconstitution.

**Nanodisc reconstitution**. MSP2N2 was prepared according to the published protocol[45]. Purified TRPV1 was concentrated to 2–2.5 mg mL$^{-1}$, and mixed with purified MSP2N2 together with a lipid mixture (1-palmitoyl-2-oleoyl-sn-glycero-3-phosphocholine (POPC), 1-palmitoyl-2-oleoyl-sn-glycero-3-phosphoethanolamine (POPE), 1-palmitoyl-2-oleoyl-sn-glycero-3-phospho-(1′-rac-glycerol) (POPG), POPC: POPE: POPG = 3:1:1; Avanti Polar Lipids) at a molar ratio of 1:3:200 (tetrameric channel:nanodisc:lipids). The channel-lipid-MSP2N2-detergent mixture was incubated at 4 °C for 45 min under gentle agitation before addition of 100 mg mL$^{-1}$ Bio-Beads SM2 (Bio-Rad). The Bio-Beads were exchanged with a fresh batch after two hours, and the mixture was incubated at 4 °C for 12–15 h with constant rocking. The mixture was then subjected to size exclusion chromatography on a Superose 6 Increase 10/300 GL column (Cytiva) pre-equilibrated with buffer containing 20 mM HEPES pH 7.5, 150 mM NaCl.

**Cryo-EM sample preparation and data collection**. Size exclusion peak fractions containing nanodisc-reconstituted TRPV1 were concentrated to 0.8–1.0 mg mL$^{-1}$. All cryo-EM samples in this study were prepared on freshly glow-discharged UltrAuFoil R1.2/1.3 300 mesh grids (Quantifoil), using a Leica EM GP2 to plunge freeze in LN2-cooled liquid ethane. (i) For TRPV1$^{4C, RTx}$, the TRPV1 sample was mixed with 50 μM RTx (Alomone, dissolved in DMSO) for 30 min before applying to the grid. Grids were blotted for 2 s at 4 °C and 90% humidity before plunge freezing. (ii) For TRPV1$^{25C, RTx}$, TRPV1 was mixed with 50 μM RTx for 15–20 min before applying to the grid, incubated at 25 °C for 30 s in a metal block, blotted for 2 s in the chamber set to 25 °C and 90% humidity, then plunge-frozen. (iii) For TRPV1$^{48C, RTx}$, TRPV1 was mixed with 50 μM RTx for 15–20 min before applying to the grid, incubated in a metal heat block set at 48 °C for 30 s, blotted for 2 s in the chamber set at 48 °C and 80% humidity, then plunge-frozen. The 30 s heat incubation time was empirically optimized to avoid channel aggregation. For 25 °C and 48 °C heat treatment experiments, the plunge freezer chamber was equilibrated to the set temperatures and all tools that contact the protein sample (including plunge freezing forceps, pipet tips, 1.5 mL tubes, and grids) were pre-heated (> 5 mins) on a heat-block.

The TRPV1$^{4C, RTx}$ dataset was collected with a Titan Krios microscope (Thermo Fisher) operating at 300 kV and equipped with a K3 detector (Gatan) in counting mode, using the Latitude-S (Gatan) automated data acquisition program. Movie datasets were collected at a nominal magnification of ×81,000 with a pixel size of 1.08 Å/pix at specimen level. Each movie contains 60 frames over a 4.6 s exposure time, using a dose rate of about 15 e$^-$/Å$^2$/s, resulting in the total accumulated dose of ~60 e$^-$/Å$^2$. The nominal defocus range was set from −0.9 to −2.0 μm.

The TRPV1$^{25C, RTx}$ dataset was collected with a Titan Krios microscope (Thermo Fisher) operating at 300 kV equipped with a K3 detector (Gatan) in counting mode, using the Serial-EM automated data acquisition program[46]. Movie datasets were collected at a nominal magnification of ×81,000 with a pixel size of 0.5145 Å/pix in super-resolution mode. Each movie contains 50 frames over a 2.4 s exposure time, using a dose rate of about 18.5 e$^-$/Å$^2$/s, resulting in the total accumulated dose of ~44 e$^-$/Å$^2$. The nominal defocus range was set from −0.8 to −1.9 μm.

The TRPV1$^{48C, RTx}$ dataset was collected with a Titan Krios microscope (Thermo Fisher) operating at 300 kV equipped with a K3 detector (Gatan) in counting mode, using the Serial-EM automated data acquisition program. Movie datasets were collected at a nominal magnification of ×81,000 with a pixel size of 0.5395 Å/pix in super-resolution mode. Each movie contains 74 frames over a 3.5 s exposure time, using a dose rate of ~ 15 e$^-$/Å$^2$/s, resulting in the total accumulated dose of ~50 e$^-$/Å$^2$. The nominal defocus range was set from −0.8 to −1.9 μm.

**Cryo-EM data processing**. All datasets were processed using similar procedures. Beam-induced motion correction and dose-weighing were performed using MotionCor2[47], followed by CTF estimation using Gctf[48]. For TRPV1$^{25C, RTx}$ and TRPV1$^{48C, RTx}$ datasets, the movies were ×2 Fourier binned during motion correction, resulting in pixel sizes of 1.029 Å/pix and 1.079 Å/pix, respectively. Micrographs were subsequently selected based on CTF fit quality and CTF estimated resolution.

**TRPV1$^{4C, RTx}$ dataset**. To obtain a consensus 3D refinement (Supplementary Fig. 1a), an initial set of particles were manually picked and subjected to a reference-free 2D classification ($k = 10$, $T = 2$), from which the best 3–5 classes were selected as references for automated particle picking in RELION 3.1[49]. Particles were extracted by 4 × 4 Fourier binning followed by a reference-free 2D classification ($k = 50$, $T = 2$) performed in RELION; classes showing clear secondary structure features of TRPV1 were selected. The selected particles were re-centered, re-extracted, 2 × 2 Fourier binned with 2.16 Å/pix and 128-pixel box size, and subjected to 3D auto-refinement to generate a 3D reference volume and find optimal orientations for subsequent 3D classification procedures in RELION, using a previously published TRPV1 map (EMD-23473, low-passed filtered to 30 Å) as reference without masking. Refined particles were subjected to 3D classification ($k = 4$, $T = 8$) with image alignment. The 337,458 particles from the classification that exhibited a well-resolved map were re-centered, re-extracted, unbinned (1.08 Å/pixel, 256-pixel box size), and subjected to 3D auto-refinement with a soft mask covering the best-resolved region of the channel, yielding a consensus map at ~3.7 Å.

To further investigate and distinguish the conformational heterogeneity within the consensus refinement, the refined particles were subjected to Bayesian polishing and CTF refinement, followed by further 3D classification ($k = 3$, $T = 10$) without image alignment using the same mask as the consensus reconstruction. The resulting three 3D classes exhibited considerable differences around the transmembrane region, especially around S6, resulting in different sizes of central pore opening and were each subjected to 3D auto-refinement. The $k$ and $T$ parameters for 3D classification were systematically tested, and the above parameters yielded the most robust and consistent results in distinguishing conformational differences within the dataset. To further refine the maps and improve resolution, the refined particles of class I (closed state) were transferred to cryoSPARC and subjected to another round of heterogenous refinement and ab initio reconstruction to remove the low-resolution particles. The selected and refined particles were finally subjected to non-uniform refinement, yielding the

class I construction of ~3.05 Å resolution determined by gold-standard 0.143 Fourier shell correlation (FSC). The class II (intermediate closed state) particles were further subjected to 3D classification ($k = 2$, $T = 20$) to remove the low-resolution particles followed by Bayesian polishing and CTF refinement. 70,786 particles from the class showing the most homogeneous conformational features were subjected to 3D auto-refinement with a full mask followed by Bayesian polishing and CTF refinement, yielding the class II reconstruction of ~3.45 Å resolution determined by RELION. The particles from class III (final-open state) were transferred to cryoSPARC for further processing and subjected to another round of heterogenous refinement and ab initio reconstruction to remove the low-resolution particles. 107,629 particles were subjected to non-uniform refinement, yielding the final class 3 reconstruction of 3.11 Å resolution determined by cryoSPARC.

**TRPV1$^{25C, RTx}$ dataset**. An initial consensus refinement (Supplementary Fig. 1b) was conducted using similar procedures. A small set of particles were manually picked and subjected to a reference-free 2D classification ($k = 10$, $T = 2$), from which the best 3 classes were selected as reference for automated particle picking in RELION 3.1. Because the total number of picked particles is below 600k, the particles were extracted by $2 \times 2$ Fourier binning with 2.058 Å/pix and 128-pixel box size instead of $4 \times 4$ Fourier binning. Reference-free 2D classification ($k = 50$, $T = 2$) was performed in RELION and classes showing clear secondary structure features of TRPV1 were selected. The selected 347,652 particles were then subjected to 3D auto-refinement to generate a 3D reference volume and find optimal orientations for subsequent 3D classification procedures. The refined particles were subjected to 3D classification with image alignment ($k = 3$, $T = 8$). The 117,332 particles that displayed the best-resolved density around the transmembrane region were re-centered, re-extracted, and unbinned (1.029 Å/pixel, 256-pixel box size), then subjected to 3D auto-refinement with a soft mask. The refined particles were subjected to 3D classification ($k = 2$, $T = 10$) without image alignment using the same mask. The $k$ and $T$ parameters for 3D classification were systematically tested, and the above parameters yielded the most robust and consistent results in distinguishing conformational differences within the dataset. The resulting two 3D classes exhibited considerable differences around the transmembrane region, especially around S6, resulting in different sizes of central pore opening. 56,119 particles were subjected to 3D-auto refinement, yielding the intermediate-open state map reconstruction of 3.73 Å resolution determined by RELION. 60,998 particles from the class showing the most homogeneous conformational features were subjected to 3D auto-refinement, yielding the final open state reconstruction of ~3.35 Å resolution determined by RELION.

For getting the better resolution structures, after the reference-free 2D classification, the selected 347,652 particles were transferred to cryoSPARC. 347,652 particles were then subjected to ab initio reconstruction with two classes, allowing the removal of junk particles. The one class showing clear channel features was retained (257,721 particles). This stack of particles was classified into three classes using heterogenous refinement with C4 symmetry, resulting in one class showing clear TRPV1 features containing 136,448 particles. Consensus refinement was generated from this stack of particles using homogenous refinement in cryoSPARC, resulting in a 3.3 Å map. To further dissect conformational heterogeneity, the consensus refined particles were transferred to RELION and subjected to Bayesian polishing and CTF refinement followed by 3D classification without image alignment ($k = 2$, $T = 10$). Both classes clearly showed proper TRPV1 features, and were subjected to 3D auto-refinement separately. After one round of Bayesian polishing and CTF refinement both classes were transferred to cryoSPARC for non-uniform refinement, yielding a 3.36 Å final map for class A (intermediate open state, 32,563 particles) and 3.04 Å for class B (final open state, 104,711 particles), respectively.

**TRPV1$^{48C, RTx}$ dataset**. An initial consensus refinement (Supplementary Fig. 1c) was conducted using similar procedures. A small set of particles were manually picked and subjected to a reference-free 2D classification ($k = 5$, $T = 2$), from which the best 2 classes were selected as reference for automated particle picking in RELION 3.1. Particles were extracted by $4 \times 4$ Fourier binning with 4.316 Å/pix and 64-pixel box size. Reference-free 2D classification ($k = 50$, $T = 2$) was performed in RELION and classes showing clear secondary structure features of TRPV1 were selected. The selected particles were then subjected to 3D auto-refinement to generate a 3D reference volume and find optimal orientations for subsequent 3D classification procedures in RELION. The refined particles were subjected to 3D classification with image alignment ($k = 3$, $T = 8$). The 214,559 particles that displayed the best-resolved density around the transmembrane region were re-centered, re-extracted, and unbinned (1.079 Å/pixel, 256-pixel box size), then subjected to 3D auto-refinement with a soft mask. The refined particles were subjected to 3D classification ($k = 4$, $T = 10$) without image alignment using the same mask. The 43,979 particles from one class that exhibited well-resolved tetrameric channel features were subjected to CTF refinement and Bayesian polishing followed by 3D classification without image alignment for improving resolution ($k = 2$, $T = 20$). In this dataset, further 3D classification did not resolve conformational heterogeneity, but converged on one high-resolution class while rejecting particles that did not have clear TRPV1 features. A single class showing the clearest and the best-resolved features (18,431 particles) was subjected to 3D

auto-refinement, yielding a final reconstruction of ~3.3 Å determined by gold-standard Fourier shell correlation (FSC) using RELION.

**TRPV1$^{4C, RTx}$ dataset for particle distribution analysis**. To alleviate bias in the subjective aspects of data processing (e.g selection of classes, etc.) and to test the robustness of our methods in ensemble generation, different 2D class selection thresholds were used during data processing. The detail of the processing strategies used here can be found in Supplementary Fig. 2a, b. After 2D classification, three sets of particles were selected where the sets progressively included particles from lower quality classes, resulting in class sizes of 411,308, 515,366 and 622,539 particles. For the 337k particle subset, particles were subjected to a 3D classification ($k = 3$, $T = 8$) with a single reference from a previously published TRPV1 map (EMD-23473) to avoid model bias. For the 411k, 515k and 622k particle subsets, particles were subjected to multiple-reference 3D classification ($k = 4$, $T = 8$) by using four reference models: three final classes from the 337k particle subset and a featureless class from a first-round classification. After each 3D classification job, the resulting classes showing clear and distinct TRPV1 conformations were selected and subjected to 3D refinement, resulting in refined maps for classes I, II and III. The numbers of particles representing each class through different classification procedures are summarized in Supplementary Fig. 2c.

**TRPV1$^{25C, RTx}$ dataset for particle distribution analysis**. Particle selection and classification followed the same strategy used for the 4 °C RTX dataset. The detail of the processing strategies used here can be found in Supplementary Fig. 2b. 2D classes showing TRPV1 features were partitioned into three particle subsets containing 257,629, 564,950 and 812,131 particles. For the 136k particle subset, particles were subjected to a 3D classification ($k = 3$, $T = 8$) with a single reference from a previously published TRPV1 map (EMD-23473) to avoid model bias. For the 257k, 564k and 812k particle subsets, particles were subjected to multi-reference heterogenous refinement (cryoSPARC) using three reference models: two final classes from the the result of 136k particle subset and one featureless class from initial classification. After each 3D classification job, the resulting classes showing clear and distinct TRPV1 conformations were selected and subjected to 3D refinement, resulting in refined maps for classes A and B. The numbers of particles representing each class through different classification procedures are summarized in Supplementary Fig. 2d.

**Model building, refinement, and validation**. The cryo-EM structure of apo TRPV1 (PDB: 7LP9) was docked into the maps using Dock in Map (as implemented in PHENIX) and manually adjusted in Coot. Model building was guided by bulky aromatic residues to ensure correct register assignment in helices and β-strands. The placement of individual elements was built by rigid body fitting and the structures were manually refined using real space refinement in Coot[50] with ideal geometric and secondary structure restraints. The restraints for lipids and ligands, including POPC, POPE, POPG, and RTx, were generated by eLbow (as implemented in PHENIX[51] from canonical SMILES strings and optimized using the REEL QM2 method (as implemented in the Phenix suite). Problematic regions identified by The MolProbity server (http://molprobity.biochem.duke.edu)[52] were adjusted in Coot manually. The final refinement was performed using Phenix-real_space_refine with global minimization, and secondary structure restrains as implemented in the Phenix suite[51]. Final models were validated by phenix_validation_cryoem with MolProbity in the Phenix suite. The Fourier shell correlation of the full- and half-maps against the models, calculated in Phenix, were in fair agreement, indicating that the models were not over-fitted nor over-refined. Structural illustrations, analysis and figure preparation were performed using UCSF Chimera[53] and PyMOL (Schrödinger). For figure generation, cryoEM density thresholds are adjusted to show the same contour level across maps. Alignments of cryo-EM maps were performed by Fit In Map in UCSF Chimera. Using the aligned maps, each structural model was re-aligned to its corresponding map.

**Patch clamp electrophysiology**. Macroscopic currents were recorded in the whole-cell configuration using transiently transfected HEK293T cells. Currents were low-pass filtered at 2 kHz (Axopatch 200B) and digitally sampled at 5–10 kHz (Digidata 1440 A), the recordings were carried out at various temperatures from ~10 °C to ~50 °C. Pipettes were pulled from borosilicate glass to final resistances of 2–5 MΩ. Electrodes were filled with an intracellular solution containing 140 mM NaCl, 5 mM MgCl$_2$, 10 mM HEPES, 5 mM EGTA, and adjusted to pH 7.4 (NaOH). The extracellular solution contained 140 mM NaCl, 10 mM HEPES, 5 mM EDTA, pH 7.4 (NaOH). Resiniferatoxin (RTx) and ruthenium red (RR) were applied using a gravity-fed perfusion system. For the temperature activation experiment, currents were recorded using a voltage ramp protocol consisting of 50 ms at a holding potential of −60 mV, 300 ms ramp to +60 mV, followed by another 50 ms at 60 mV, which was applied every 500 ms for the duration of the recording. Data collected at −60 mV were used for analyzing channel behavior. The external recording buffer was passed through glass capillary coils immersed in a hot water bath maintained at ~80 °C and then an ice-water bath for quick cooling to reduce cell damage. The temperature was measured by a thermistor (TA-29, Warner Instruments) placed very close to the pipette tip. For measuring RTx-bound TRPV1 temperature sensitivity, we first perfused with various low concentrations

of RTx (2–10 nM) for various time increments (5–30 s) to achieve stable partial activation of TRPV1, followed by exchange to RTx-free solutions to stop further activation of TRPV1. This nearly irreversible binding of RTx induces TRPV1 opening for extended periods of time, even after exchange to RTx-free solution. We then applied a temperature ramp (from ~10 °C to 40 °C–50 °C) followed by application of a saturating concentration of RTx (50 nM). The apparent open probability induced by RTx was determined by $I_{intial}/I_{50nM\ RTx}$. For $Q_{10}$ calculation, we analyzed the Arrhenius plot for each recording and applied the equation $Q_{10} = 10^{\wedge}(-10 \times S_{Arrhe})/(T_1 \times T_2)$ to calculate $Q_{10}$ values for RTx free or RTx-bound TRPV1. $S_{Arrhe}$ is the slope of the linear fit to Arrhenius plotted data between absolute temperatures $T_1$ and $T_2$. For inside-out patch, TRPV1 channels were activated by focal perfusion of 200 nM RTx, followed by application of 10 μM YO-PRO-1 to the cytoplasmic side of the excised patches. A repeated ramp protocol from −120 mV to +120 mV was used to elicit channel activity with intervening 2s intervals where the membrane was held at 0 mV. Peak currents at +120 mV were used for data analysis.

**Thermodynamic treatment of particle distribution analysis**. A simple 3-state model was used to estimate equilibrium constants between non-conducting closed and intermediate states as well as the fully conducting open state. Equilibria were modeled using the proportions of particles in each state from the particle distribution analysis. Equilibrium constants $K_1$ and $K_2$ represent Closed→Intermediate and Intermediate→Open, respectively. Enthalpy values for the temperature-dependent shifts in equilibria were assumed constant (i.e. not factoring in heat capacity change) and calculated using the equation:

$$ln\left(\frac{K_B}{K_A}\right) = \frac{\triangle H}{R}\left(\frac{1}{T_A} - \frac{1}{T_B}\right) \qquad (1)$$

In cases where a class of particle was not observed in the dataset, a population value of 1 was used in order to roughly estimate thermodynamic parameters. $Q_{10}$ values were calculated considering the equilibrium observed in electrophysiology experiments as: (3) $p_{Open}/(p_{Closed} + p_{Intermediate})$. Respective apparent $Q_{10}$ values were then approximated by $Q_{10} \approx e^{(\triangle H/20)}$.

**Reporting summary**. Further information on research design is available in the Nature Research Reporting Summary linked to this article.

## Data availability

The coordinates generated in this study are deposited in the Protein Data Bank with the PDB IDs 7RQU (TRPV1[4C,RTx,closed]), 7RQV (TRPV1[4C,RTx,Inter]), 7RQW (TRPV1[4C,RTx,open]), 7RQX (TRPV1[25C,RTx,Inter]), 7RQY (TRPV1[25C,RTx,open]) and 7RQZ (TRPV1[48C,RTx,open]), respectively. The cryo-EM maps are deposited in the Electron Microscopy Data Bank with the IDs EMD-24636 (TRPV1[4C,RTx,closed]), EMD-24637 (TRPV1[4C,RTx,Inter]), EMD-24638 (TRPV1[4C,RTx,open]), EMD-24639 (TRPV1[25C,RTx,Inter]), EMD-24640 (TRPV1[25C,RTx,open]), EMD-24641 (TRPV1[48C,RTx,open]), respectively. Source data for Figs. 1 and 6 are provided.

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

## Acknowledgements
Cryo-EM data were collected at the Duke University Shared Materials Instrumentation Facility (SMIF) and at the Pacific Northwest Center for Cryo-EM (PNCC) at OHSU. We thank Janette Myers at PNCC for assistance in data collection and Nilakshee Bhattacharya at SMIF for assistance with the microscope operation. This research was supported by a NIH grant (R35NS097241 to S.-Y.L.). A portion of this research was supported by NIH grant U24GM129547 and performed at the PNCC at OHSU and accessed through EMSL (grid.436923.9), a DOE Office of Science User Facility sponsored by the Office of Biological and Environmental Research. DUKE SMIF is affiliated with the North Carolina Research Triangle Nanotechnology Network, which is in part supported by the NSF (ECCS-2025064).

## Author contributions
D.K. conducted biochemical preparation, sample freezing, single-particle 3D reconstruction, and model building under the guidance of S.-Y.L. F.Z. carried out all electrophysiological recordings under the guidance of S.-Y.L. Y.S. collected part of the cryo-EM data and helped D.K. for part of data processing. J.G.F. performed the thermodynamic analysis. S.-Y.L., D.K., and J.G.F. wrote the paper.

## Competing interests
The authors declare no competing interests.
