## [Peer Review File · Nature Communications]

Vanilloid-dependent TRPV1 opening trajectory from cryoEM ensemble analysisREVIEWER COMMENTS

Reviewer #1 (Remarks to the Author):

The TRPV1 ion channel is a key sensory detector of noxious stimuli and inflammatory mediators in nociceptive neurons. The TRPV1 channel integrates physical and chemical information through complex allosteric mechanisms, where interaction of the channel with an activating stimulus leads to pore opening and a net inward flow of cations into the neuron and also sensitizes responses of the channel to other stimuli. Our current understanding of how the binding of any stimulus to TRPV1 leads to opening of the pore of the channel or sensitization to other stimuli is limited, despite the recent onslaught of structural data for TRPV1 obtained under multiple conditions. One important limitation of this available data is that no open structures of TRPV1 have been observed for channels in the presence of a single activating stimulus. In the present manuscript, Do Hoon Kwon and collaborators determine structures of full-length rat TRPV1 channels in nanodiscs at three different temperatures and in the presence of a saturating concentration of the potent agonist resiniferatoxin (RTx). The authors observe three separate structural classes that are interpreted as representing closed, intermediate-open and open functional states based on the diameter of the pore. Notably, the open state observed at 4C is very similar to the predominant state observed at 48C, suggesting that it represents the open state that becomes stabilized by the binding of RTx in the absence of any other activating stimuli. Although this observation is novel and potentially relevant, I consider that the interpretation of the data is largely based on unsupported assumptions and the conclusions are also not strongly supported by the experimental evidence that is provided. These are my concerns:

1) A key assumption of this study is that the diameter of the pore constitutes a good reaction coordinate to mechanistically establish a sequence of events leading to activation, which is not justified. This is because TRPV1 channels can open in the absence of stimulation (albeit with low P_o) and channels fully liganded by RTx still undergo closure events. Without knowledge of the energetics associated with the observed conformational changes in the pore and other regions of the protein, it is impossible to confidently arrange a given set of structures along a defined mechanistic sequence of events towards activation only based on the diameter of the pore.

2) Channel behavior in biochemical preparations used for structural determination is very different from that of channels expressed in the membrane of a living cell. If the occupancies of each of the observed conformational states in the biochemical preparations closely reflected the distribution of functional states observed in electrophysiology experiments, this would provide a stronger support for the proposed sequence of events and mechanism. However, the biochemical preparations exhibit undeniable and robust discrepancies relative to the behavior of TRPV1 channels expressed in cellular membranes. In cells RTx functions as a full agonist at room temperature and likely also does at colder temperatures, because a weaker agonist like capsaicin still produces significant channel activation at very cold temperatures (see PMID: 30038260 and PMID: 26882503). However, in the biochemical

preparations at room temperature, only 50% of observed particles occupy an open state and at 4C only 30% of the channels are open, whereas 30% occupy a state that is very similar to apo. This strongly indicates that the conditions required for structural determination alter the energetics that determine channel behavior. It is therefore a possibility that some of the observed structural states are only minimally occupied when channels are in a living cell, and are therefore not representative of key conformational changes leading to activation. This key limitation is not reflected in the presentation of the results in the manuscript.

3) The argument that RTx binding ablates temperature sensitivity is not justified. When in a cell membrane, TRPV1 channels are potently activated by RTx, such that a fully liganded channel has an open probability (P_o) close to 1. This would leave a negligible dynamic range to observe a further increase in P_o upon heating. Macroscopic current recordings such as in Figure 1a reflect current responses arising from both liganded and unliganded channels (the authors themselves propose that their quantified ratio reflects RTx occupancy). Without knowing the open probability at different stoichiometries of RTx binding, it is impossible to determine whether RTx-bound channels indeed have reduced temperature sensitivity. For capsaicin, a much weaker agonist than RTx, it has been proposed that single-subunit occupancy leads to maximal activation (PMID: 26194846). Therefore, it is likely that the steep temperature-dependence observed at lower RTx concentrations, where occupancy as quantified by the authors is <20%, arises from unliganded channels, whereas liganded channels are already fully activated and therefore would exhibit no apparent temperature sensitivity even if the temperature sensor still became activated upon heating. No information is therefore provided regarding the state of the heat-sensor for RTx-bound channels. An alternative interpretation that is as likely as the one suggested by the authors is that RTx binding reduces the barrier for temperature sensor activation, and some of the structural changes that are observed reflect heat-driven and not RTx-driven conformations. None of these extremely relevant nuances are considered in the manuscript.

Further, the authors argue in the Discussion that capsaicin-bound channels retain heat-sensitivity whereas RTx-bound channels do not. However, experimental data shows that in saturating concentrations of capsaicin TRPV1 channels exhibit limited heat-sensitivity because they are already fully activated (see PMID: 30038260 and PMID: 26882503), and strongly suggest that the heat-sensitivity observed in macroscopic current recordings (i.e. in large ensembles of channels) arising from TRPV1 channels in the presence of lower capsaicin concentrations arises from either partially occupied or unoccupied channels, which have open probabilities significantly lower than 1. It is therefore not possible to ascertain the state of the heat sensor in channels fully occupied by capsaicin or RTx, and no functional or structural information is provided in the study regarding partially occupied channels, which are the ones where heat sensitivity would be detectable.

4) A central concern for this work is that almost no actual experimental evidence is provided in support of the conclusions, because none of the relevant figures show the EM densities and instead only the fitted models are shown. It is concerning that most of the key conformational differences discussed that are observed between the different states are smaller than the resolution associated with the

experimental maps (Fig. 2a-d, Fig. 3a, b and c, Fig. 4, Fig. 5, Suppl. Fig. 6, Suppl. Fig. 7, Suppl. Fig. 8, Suppl. Fig. 9, Suppl. Fig. 10). The fragmented densities in Suppl. Fig. 3 are not useful for assessing the robustness of any of the conformational changes suggested by the authors, and indeed the RMSD between different structural models are in general much smaller than the resolution for each of the corresponding maps, raising concerns on the significance of the observations. Densities are only displayed in Fig. 3c, where it is argued that the density for one side-chain conformation has higher occupancy in one state relative to the other. The difference seems to be minimal and of uncertain significance, as no method of quantitation is provided. Importantly, no densities for that same position are shown for the apo or the open states, which is essential to confidently establish a structural difference between the states at that position.

5) No additional evidence from channel function is provided to support key conclusions. For example, the state-dependent interaction between the three residues in Fig. 5 would be straightforward to test experimentally. The previous observation that a mutation at T641 disrupts permeation of large organic cations further suggests a role of that position in establishing permeation and selectivity properties of open channels, rather than being required as a key step towards activation, as proposed in this manuscript. That the alanine mutation at Y666 is non-functional does not constitute satisfactory evidence for the proposed mechanism either.

6) Finally, no information or evidence is provided to support the claim that a near complete activation pathway was identified in the present study. Indeed, others have found additional conformations in the presence of RTx that were not observed in this manuscript (PMID: 34496225), and there is no evidence that the the observed conformations, favored by the conditions of the biochemical preparation, reflect a comprehensive sampling of the functionally relevant conformational ensemble associated with TRPV1 channel activation. Importantly, without information on all the relevant conformational states, and their energetic relations, including partially ligated states that are not considered in this manuscript, it is formally impossible to draw any conclusions regarding the cooperativity associated with channel activation by the ligand, or the nature of the 'conformational wave' that could potentially be associated with it.

Reviewer #2 (Remarks to the Author):

General remarks:

The manuscript (NCOMMS-22-52084) "The conformational trajectory of vanilloid-dependent TRPV1 opening revealed through cryoEM ensembles" by Do Hoon Kwon et al. presents cryo-EM structures of vanilloid-dependent TRPV1 captured at different temperatures. The authors use "thermal titration

methods”, a temperature-dependent conformation equilibrium in a single dataset to dissect the conformational trajectory of resiniferatoxin (RTx) activated TRPV1 ion channel. They concluded that RTx binding to TRPV1 induces intracellular gate opening first, followed by selectivity filter dilation, then pore loop rearrangement to reach the final open state. This apparent conformational wave arises from the concerted, stepwise, additive structural changes of TRPV1 over many subdomains. The design of the experiment is novel and elegant, the data is solid, and the conclusions are convincing. However, I have some concerns about the bias possibly introduced during the image processing, especially the 3D classifications (see below for more details).

Major concerns:

1. In Cryo-EM data processing section, the author used K=3 in TRPV14C, TRx dataset when doing second round of 3D classification (line 422). How did this K value of 3 be selected? What if you classified the data into more than 3 classes? There might be more intermediate conformational states in the dataset can be identified, or multiple 3D classes with same structure with larger K value. In later case it would confirm that there are only 3 distinct conformational states in the dataset.
2. Same concerns about the TRPV125C, RTx dataset. How did the K value of 3 (line 4490) and 2 (line 452) be selected? Why didn't authors use the larger K values?

Minor questions/suggestions:

1. How to distinguish between thermal energy driven shift of the population of conformational states and the activation of heat sensor? (lines 118-120)
2. Line 127-8: The 3 different pore opening was due to variation in temperature, why “they may represent different conformational states along the ligand-dependent activation pathway”?
3. Line 234: “those to” might be “to those”?
4. Line 317:as well as being sensitive to the protein construct. What is the experimental result to support this notion?
5. Why the 3 data sets were collected at different settings (imaging mode, number of frames in a movie stack, pixels, electron dose rate, exposure time, defocus range, et al)?
6. How to map the 3D reconstructions of different state of RTx-TRPV1 complex to individual particles on micrographs, as show on Fig. 1d?
7. Lines 482-483: Why the 3D auto refinement was performed before 3D classification?
8. Lines 518: the initial 136 k particle were classified into 3 classes, but only two classes were shown on supplementary Fig.2d. How about the 3rd classes?

9. Line 521: what does “initial classification” refer to?

10. In Cryo-EM data processing section, there are a lot of discrepancies among datasets. For example, some are extracted with 4x binning (line 411, 480), while other with 2x binnings (line 445); some data were exported to cryoSPARC for higher resolution, while others not. Please explain the reason for doing so.

Reviewer #4 (Remarks to the Author):

This manuscript from Kwon and colleagues describes the structures of nearly complete conformational trajectory of TRPV1, providing new insights into the dynamics of ligand-mediated conformational changes. The presented structures are the first fully conducting structure for TRPV1, and set up a universal approach for studying temperature-sensitive TRPs, paving the way for further functional studies. This is an elegant and interesting work. I have only a few suggestions for enhancing the presentation of the data, which the authors might consider.

Comments:

(1) Lines 84-86, “despite the fact that in 84 electrophysiology studies RTx alone elicits channel open probabilities reaching 1”, this should be clearly stated, together with the meaning of “reaching 1”.

(2) In the Discussion part, the authors of the current manuscript should expand and discuss how their data compares, aligns any differences with the published data and clarify the most important and novel points of this manuscript.

(3) In “Cryo-EM sample preparation and data collection” of the Methods part, the authors should describe more details about how to keep the samples at constant temperatures as this manuscript may set up a new method for further functional and structural study of temperature-sensitive TRPs. For example, how to ensure the samples can be kept in the specific temperature. Why are the samples incubated in a heat block just for 30 s but not incubated longer time?

REVIEWER COMMENTS

Reviewer #1 (Remarks to the Author):

The TRPV1 ion channel is a key sensory detector of noxious stimuli and inflammatory mediators in nociceptive neurons. The TRPV1 channel integrates physical and chemical information through complex allosteric mechanisms, where interaction of the channel with an activating stimulus leads to pore opening and a net inward flow of cations into the neuron and also sensitizes responses of the channel to other stimuli. Our current understanding of how the binding of any stimulus to TRPV1 leads to opening of the pore of the channel or sensitization to other stimuli is limited, despite the recent onslaught of structural data for TRPV1 obtained under multiple conditions. One important limitation of this available data is that no open structures of TRPV1 have been observed for channels in the presence of a single activating stimulus. In the present manuscript, Do Hoon Kwon and collaborators determine structures of full-length rat TRPV1 channels in nanodiscs at three different temperatures and in the presence of a saturating concentration of the potent agonist resiniferatoxin (RTx). The authors observe three separate structural classes that are interpreted as representing closed, intermediate-open and open functional states based on the diameter of the pore. Notably, the open state observed at 4°C is very similar to the predominant state observed at 48°C, suggesting that it represents the open state that becomes stabilized by the binding of RTx in the absence of any other activating stimuli. Although this observation is novel and potentially relevant, I consider that the interpretation of the data is largely based on unsupported assumptions and the conclusions are also not strongly supported by the experimental evidence that is provided. These are my concerns:

1) A key assumption of this study is that the diameter of the pore constitutes a good reaction coordinate to mechanistically establish a sequence of events leading to activation, which is not justified. This is because TRPV1 channels can open in the absence of stimulation (albeit with low P_o) and channels fully liganded by RTx still undergo closure events. Without knowledge of the energetics associated with the observed conformational changes in the pore and other regions of the protein, it is impossible to confidently arrange a given set of structures along a defined mechanistic sequence of events towards activation only based on the diameter of the pore.

→ Thank you for your comment. We would like to point out that our presented approach of using population shifts using the thermal titration method does not need to invoke the pore diameter consideration for determining the conformational sequence of events. We observed three states (I, II, and III) at 4°C, two states (II/A and III/B) at 25°C, and one state (III/class α) at 48°C with a temperature dependence of $Q_{10} \sim 3$. Therefore, without the pore diameter consideration, we know that conformational sequence is from the class I to class II to class III. As for the further dissection of the class 2s (intermediate states), we had utilized our observation that the methionine flipped state in an intermediate open state is conformationally closer to the class 3 (the final open state). We introduced the analysis of pore diameter and domain movements in our studies to further the mechanistic understanding of TRPV1 gating. We clarified this aspect of our strategy in the discussion (line 394-398).

2) Channel behavior in biochemical preparations used for structural determination is very different from that of channels expressed in the membrane of a living cell. If the occupancies of each of the observed conformational states in the biochemical preparations closely reflected the distribution of functional states observed in electrophysiology experiments, this would provide a stronger support for the proposed sequence of events and mechanism. However, the biochemical preparations exhibit undeniable and robust discrepancies relative to the behavior of TRPV1 channels expressed in cellular membranes. In cells RTx functions as a full agonist at room temperature and likely also does at colder temperatures, because a weaker agonist like capsaicin still produces significant channel

activation at very cold temperatures (see PMID: 30038260 and PMID: 26882503). However, in the biochemical preparations at room temperature, only 50% of observed particles occupy an open state and at 4°C only 30% of the channels are open, whereas 30% occupy a state that is very similar to apo. This strongly indicates that the conditions required for structural determination alter the energetics that determine channel behavior. It is therefore a possibility that some of the observed structural states are only minimally occupied when channels are in a living cell, and are therefore not representative of key conformational changes leading to activation. This key limitation is not reflected in the presentation of the results in the manuscript.

→ We would like to point out that significantly more than 50% of channels are open at RT, as only intermediate and fully open states exist in our classification. Because we cannot distinguish intermediate closed and intermediate open states, we cannot accurately assign the structurally observed open probability, but assuming the equal populations between the two, the open probability is ~75% at RT, suggesting that our structurally-based state distribution is not too far from those in the cell membranes. Regardless, we fully agree that there is always some deviation between *in vitro* and *in vivo*, and recombinant conditions, not the least of which is the absence of a transmembrane potential for membrane proteins and other effects of the membrane. Consequently, the biochemical conditions for structure determination are often not correlated with *in vivo* experiments.

Many previous cryo-EM studies of TRPV1 (PMID: 24305160, 27281200, 34496225) cannot explain how a single ligand activates the TRPV1 channel, the novelty and utility of this study is in filling the gap between *in vitro* and *in vivo* observations. Prior cryo-EM studies of TRPV1 have suggested that at least two ligands are required to fully open the channel *in vitro*. Since RTx alone can fully activate TRPV1 *in vivo*, these structure-based conclusions give inaccurate impressions about TRPV1 function. Our study demonstrates that a single ligand can fully activate TRPV1 *in vitro*, same as *in vivo*, and is therefore the most physiologically relevant and accurate cryoEM study of RTx-TRPV1 to date. Naturally, minor states may be missed in our analysis, or exist at levels too low for detection, and of course these states could be altered relative to those of TRPV1 in the cell. These caveats have been addressed in the revised manuscript (lines 456-464).

3) The argument that RTx binding ablates temperature sensitivity is not justified. When in a cell membrane, TRPV1 channels are potentially activated by RTx, such that a fully liganded channel has an open probability (P_o) close to 1. This would leave a negligible dynamic range to observe a further increase in P_o upon heating. Macroscopic current recordings such as in Figure 1a reflect current responses arising from both liganded and unliganded channels (the authors themselves propose that their quantified ratio reflects RTx occupancy). Without knowing the open probability at different stoichiometries of RTx binding, it is impossible to determine whether RTx-bound channels indeed have reduced temperature sensitivity. For capsaicin, a much weaker agonist than RTx, it has been proposed that single-subunit occupancy leads to maximal activation (PMID: 26194846). Therefore, it is likely that the steep temperature-dependence observed at lower RTx concentrations, where occupancy as quantified by the authors is <20%, arises from unliganded channels, whereas liganded channels are already fully activated and therefore would exhibit no apparent temperature sensitivity even if the temperature sensor still became activated upon heating. No information is therefore provided regarding the state of the heat-sensor for RTx-bound channels. An alternative interpretation that is as likely as the one suggested by the authors is that RTx binding reduces the barrier for temperature sensor activation, and some of the structural changes that are observed reflect heat-driven and not RTx-driven conformations. None of these extremely relevant nuances are considered in the manuscript.

Further, the authors argue in the Discussion that capsaicin-bound channels retain heat-sensitivity whereas RTx-bound channels do not. However, experimental data shows that in saturating

concentrations of capsaicin TRPV1 channels exhibit limited heat-sensitivity because they are already fully activated (see PMID: 30038260 and PMID: 26882503), and strongly suggest that the heat-sensitivity observed in macroscopic current recordings (i.e. in large ensembles of channels) arising from TRPV1 channels in the presence of lower capsaicin concentrations arises from either partially occupied or unoccupied channels, which have open probabilities significantly lower than 1. It is therefore not possible to ascertain the state of the heat sensor is channels fully occupied by capsaicin or RTx, and no functional or structural information is provided in the study regarding partially occupied channels, which are the ones where heat sensitivity would be detectable.

→ These are good points, so we thank the reviewer for bringing this up. It is possible that either RTx binding ablates temperature sensitivity or fully RTx bound channel is constitutively open (at RT), masking high heat dependent opening.

Our structural observation is in line with the first possibility. First, we would like to point out that the logP for RTx is 4.5 (PubChem XLogP3-AA), indicating high effective concentrations of RTx can be reached in the membrane even at low bulk concentrations of RTx. This is similar to capsaicin where the reported partition coefficient is 3×10^5 (PMID: 34239123). With cryoEM, we can observe the pore diameter and therefore ascertain the degree of channel opening. In our structure, RTx appears to saturate all four binding sites, thanks to the high concentration of RTx (50 μ M) used for sample preparation. However, we do still observe closed, intermediate-closed, intermediate-open and fully open classes, with population shifts following general heat-dependent thermodynamics (low Q_{10} values). If the RTx-bound TRPV1 had retained heat sensitivity but with a significantly lowered threshold, we would expect to see a much more pronounced temperature-dependent shift in class populations. Importantly, the conformation of the RTx-bound fully open state at 4°C is similar to that of the RTx-bound fully open state at 48°C, strongly suggesting that the fully open state conformation is due to ligand gating. The caveat here is that out of the membrane, the nanodisc-reconstituted channel may generally have altered heat sensitivity. Last, we feel that discussion of partial occupancy and its effects, while important, is beyond the scope of this work. To address this point in an unbiased way, we have included it among the caveats of our work in the discussion (lines 120-122, 452-464).

4) A central concern for this work is that almost no actual experimental evidence is provided in support of the conclusions, because none of the relevant figures show the EM densities and instead only the fitted models are shown. It is concerning that most of the key conformational differences discussed that are observed between the different states are smaller than the resolution associated with the experimental maps (Fig. 2a-d, Fig. 3a, b and c, Fig. 4, Fig. 5, Suppl. Fig. 6, Suppl. Fig. 7, Suppl. Fig. 8, Suppl. Fig. 9, Suppl. Fig. 10). The fragmented densities in Suppl. Fig. 3 are not useful for assessing the robustness of any of the conformational changes suggested by the authors, and indeed the RMSD between different structural models are in general much smaller than the resolution for each of the corresponding maps, raising concerns on the significance of the observations. Densities are only displayed in Fig. 3c, where it is argued that the density for one side-chain conformation has higher occupancy in one state relative to the other. The difference seems to be minimal and of uncertain significance, as no method of quantitation is provided. Importantly, no densities for that same position are shown for the apo or the open states, which is essential to confidently establish a structural difference between the states at that position.

→ Thank you for your suggestion. We have prepared a new figure showing the cryo-EM maps and coordinates around the selectivity filter and S6 gate, thus providing more robust support for our conclusions (Figure. 4).

We would like to point out that local resolutions of the selectivity filter and S6 gate are higher than the overall resolution (Fig. S1). So, while the global RMSD might seem small, local structural difference is large enough to distinguish the conformational changes without ambiguity.

Please see our maps. Most importantly, our reconstructions have been very robust and consistent, having collected numerous TRPV1 data which are all in excellent agreement across conditions (PMID: 34239123).

5) No additional evidence from channel function is provided to support key conclusions. For example, the state-dependent interaction between the three residues in Fig. 5 would be straightforward to test experimentally. The previous observation that a mutation at T641 disrupts permeation of large organic cations further suggests a role of that position in establishing permeation and selectivity properties of open channels, rather than being required as a key step towards activation, as proposed in this manuscript. That the alanine mutation at Y666 is non-functional does not constitute satisfactory evidence for the proposed mechanism either.

→ Thank you for your comment. In response to the reviewer's request, we tested the effects of the point mutations Y584F and T641A on the permeation of YO-PRO-1, on the RTX-induced Na⁺ currents, and on the reduction of Na⁺ currents by YO-PRO-1. We found that Y666F or Y666A exhibited no appreciable currents, which makes the state-dependent studies difficult. The uptake of YO-PRO-1 by the Y584F and T641A mutants are significantly reduced and slowed, while its apparent Na⁺ conductance was preserved (Figure 6b-e). The similar mutational effect of Y584F on S5 and T641 (pore helix) is consistent with the hypothesis that these residues form a defined and mechanistically important interaction network. The results of these experiments have been included in the revised version of the manuscript (Figure 6b-e) (Lines 292-300). In addition, our conformational trajectory by cryo-EM is consistent with the conformational wave studies done by functional studies (Φ analysis) by the Jie Zheng's group, which we have cited and clarified in the text (line 254-262).

6) Finally, no information or evidence is provided to support the claim that a near complete activation pathway was identified in the present study. Indeed, others have found additional conformations in the presence of RTX that were not observed in this manuscript (PMID: 34496225), and there is no evidence that the observed conformations, favored by the conditions of the biochemical preparation, reflect a comprehensive sampling of the functionally relevant conformational ensemble associated with TRPV1 channel activation. Importantly, without information on all the relevant conformational states, and their energetic relations, including partially ligated states that are not considered in this manuscript, it is formally impossible to draw any conclusions regarding the cooperativity associated with channel activation by the ligand, or the nature of the 'conformational wave' that could potentially be associated with it.

→ Thank you for your comment. Regarding the additional conformations that were reported but not observed in this study, there may be several reasons why we did not observe partially ligated states as in other studies, which may stem from discrepancies in the biochemical conditions, such as expression constructs (N-terminus MBP fusion construct vs. C-terminus flag construct), ligand incubation times (20 mins vs. > 20 mins) and concentration (40 μ M RTX vs. 50 μ M RTX). Also, there is undoubtedly more intermediate states than what we observed, as protein dynamics lie along a continuum. It is, however, impossible to capture all different states due to current cryo-EM technical limitations, which is why we refer to it as a near complete activation pathway. The important point that we would like to make is that it is possible to capture multiple different conformations with different biochemical conditions. Whether those conformations represent on-pathway conformations and, if so, where are they situated in the conformational trajectory, are big questions. What is clear is that the multiple states we observe here can be mapped to a conformational continuum for TRPV1 gating by a ligand, as evidenced by the progressive changes in conformations of various subdomains and the pore, quantified by relative RMSD between states. Most importantly, using one ligand (single modality), we observe multiple conformations sampling the trajectory from closed to open channel.

Reviewer #2 (Remarks to the Author):

General remarks:

The manuscript (NCOMMS-22-52084) “The conformational trajectory of vanilloid-dependent TRPV1 opening revealed through cryoEM ensembles” by Do Hoon Kwon et al. presents cryo-EM structures of vanilloid-dependent TRPV1 captured at different temperatures. The authors use “thermal titration methods”, a temperature-dependent conformation equilibrium in a single dataset to dissect the conformational trajectory of resiniferatoxin (RTx) activated TRPV1 ion channel. They concluded that RTx binding to TRPV1 induces intracellular gate opening first, followed by selectivity filter dilation, then pore loop rearrangement to reach the final open state. This apparent conformational wave arises from the concerted, stepwise, additive structural changes of TRPV1 over many subdomains. The design of the experiment is novel and elegant, the data is solid, and the conclusions are convincing. However, I have some concerns about the bias possibly introduced during the image processing, especially the 3D classifications (see below for more details).

Major concerns:

1. In Cryo-EM data processing section, the author used K=3 in TRPV1 4C, TRx dataset when doing second round of 3D classification (line 422). How did this K value of 3 be selected? What if you classified the data into more than 3 classes? There might be more intermediate conformational states in the dataset can be identified, or multiple 3D classes with same structure with larger K value. In later case it would confirm that there are only 3 distinct conformational states in the dataset.

→ Thank you for this suggestion. However, as seen in Fig. S2, we did some tests to determine whether there are more distinct conformational states by varying a couple of different parameters [(number of particles and different number of classes (3 or 4)). Also, when we tested 3D classification with a K value of 5, only three resulting classes exhibit TRPV1 channel features whereas the other two classes were “junk” classes without discernible features (see below). There may be additional subtly different intermediate states, but we believe they cannot be robustly separated due to the current limitations in cryo-EM data processing algorithms.

4°C dataset (337K particles)

Single reference 3D classification K=5

2. Same concerns about the TRPV125C, RTx dataset. How did the K value of 3 (line 4490) and 2 (line 452) be selected? Why didn't authors use the larger K values?

→ We also tried the 3D classification with a K value of 4, but were unable to find additional intermediate states in this data set (see below). We included the following sentence to clarify our systematic approach: "The k and T parameters for 3D classification were systematically tested, and the above parameters yielded the most robust and consistent results in distinguishing conformational differences within the dataset." (lines 618-620, 650-652)

25°C dataset

Single reference 3D classification K=4

Minor questions/suggestions:

1. How to distinguish between thermal energy driven shift of the population of conformational states and the activation of heat sensor? (lines 118-120)

→ As we mentioned and have further clarified in the text (lines 115-122), we measured the heat sensitivity (Q_{10}) of RTx-bound TRPV1 at high and low temperatures, with Q_{10} values below 3. Such low Q_{10} values indicate a reduction in heat sensitivity for the RTx-bound TRPV1 to the level of canonical ion channel gating. In conclusion, this population shift is driven purely by general thermal energy effects and is not due to a particular thermosensory mechanism of heat-activation (where $Q_{10} \sim 20-30$).

2. Line 127-8: The 3 different pore opening was due to variation in temperature, why “they may represent different conformational states along the ligand-dependent activation pathway”?

→ We apologize for the confusion. Lines 127-128 described how observing 3 classes at the same temperature (4°C) led us to hypothesize that the different pore openings represent different conformational states along the ligand-dependent activation pathway. We had included this sentence for contextual flow and have modified it to explicitly state that it is referring to the 4°C dataset (Line 141). Data collected at higher temperatures resulted in classes with wider pore opening and are nearly identical to the second and the third classes of the 4°C dataset. Because conformations of various pore openings exist in the 4°C dataset, and the temperature increase only gradually increases the population of the states with wider pore openings, we concluded that observed pore opening is ligand dependent. As mentioned above, Q_{10} values below 3 indicate the loss of specific temperature sensitivity by RTx-bound TRPV1, with thermal sensitivity due to general temperature effects. Therefore, pore gating in this system depends entirely on the ligand.

3. Line 234: “those to” might be “to those”?

→ Thank you for pointing this out, we have changed it (line 253).

4. Line 317:as well as being sensitive to the protein construct. What is the experimental result to support this notion?

→ This is based on the previous structural study by the Cheng and Julius groups which showed that only truncated TRPV1 can be activated by a single ligand. However, since we only used the full-length construct and thus do not have our own data on this point, we have removed our speculation about the protein construct in the text. (removed)

5. Why the 3 data sets were collected at different settings (imaging mode, number of frames in a movie stack, pixels, electron dose rate, exposure time, defocus range, et al)?

→ We agree that it is ideal to collect all 3 datasets using the same microscope with the same imaging settings. Unfortunately, limited cryo-EM resources and time availability make it impractical to collect all datasets on a single microscope, therefore we had to apply for data collections at multiple national facilities. Because each national facility has a preferred setting for each microscope, there is a small difference in imaging modes. We have collected numerous TRPV1 data at various temperatures at different facilities and the resulting reconstructions are always very consistent (PMID 34239123). We are confident that these setting do not affect our analysis and conclusion.

6. How to map the 3D reconstructions of different state of RTx-TRPV1 complex to individual particles on micrographs, as show on Fig. 1d?

→ During particle picking, the particle coordinates are recorded in a particle metadata file, which is preserved through subsequent classification and refinement. After the final refinement and identification of different functional states, we extracted the particle coordinates of each state and re-mapped them onto the original micrographs.

7. Lines 482-483: Why the 3D auto refinement was performed before 3D classification?

→ A 3D reference model, as well as initial angular/translation assignment is required as input (a Bayesian prior) to 3D classification. In our experience, it is helpful to generate the 3D reference from the dataset itself to avoid overfitting and model bias. Moreover, we perform this consensus 3D refinement to see if the data is of good quality before proceeding to dissect more subtle differences within the dataset.

8. Lines 518: the initial 136 k particle were classified into 3 classes, but only two classes were shown on supplementary Fig.2d. How about the 3rd classes?

→ We thank the reviewer for rightly pointing this out. We have updated Fig.S2 to show this, the 3rd class being “junk” without identifiable TRPV1 protein features.

9. Line 521: what does “initial classification” refer to?

→ We are sorry for the confusion. We are simply referring to the first round of 3D classification. We have edited the text accordingly (line 709).

10. In Cryo-EM data processing section, there are a lot of discrepancies among datasets. For example, some are extracted with 4x binning (line 411, 480), while other with 2x binnings (line 445); some data were exported to cryoSPARC for higher resolution, while others not. Please explain the reason for doing so.

→ We typically extract particles with 4x binning at an early stage to expedite processing. In the case of the 25C dataset, because there were few selected particles in the initial 2D classification, we extracted particles with 2x binning. In our experience with TRPV1 data, 4X or 2X binning does not make any difference in final reconstructions. We have carefully assessed these points and used the same software for all classifications to ensure robust results in the classification and the relative populations. We have only changed the software to obtain the highest resolution and quality maps. (lines 583-584).

Reviewer #4 (Remarks to the Author):

This manuscript from Kwon and colleagues describes the structures of nearly complete conformational trajectory of TRPV1, providing new insights into the dynamics of ligand-mediated conformational changes. The presented structures are the first fully conducting structure for TRPV1, and set up a universal approach for studying temperature-sensitive TRPs, paving the way for further functional studies. This is an elegant and interesting work. I have only a few suggestions for

enhancing the presentation of the data, which the authors might consider.

Comments:

(1) Lines 84-86, “despite the fact that in 84 electrophysiology studies RTx alone elicits channel open probabilities reaching 1”, this should be clearly stated, together with the meaning of “reaching 1”.

→ Thank you for drawing this need for clarification to our attention. We have modified the text accordingly, from:

“This led to the belief that single modality activation of TRPV1 could not be achieved²⁵, despite the fact that in electrophysiology studies RTx alone elicits channel open probabilities reaching 1 and allows for conduction of large organic cations^{26,27}”

To:

“This led to the belief that single modality activation of TRPV1 could not be achieved²⁵, despite the fact that in electrophysiology studies RTx alone elicits channel open probabilities reaching 1 and allows for conduction of large organic cations^{26,27}.”

(2) In the Discussion part, the authors of the current manuscript should expand and discuss how their data compares, aligns any differences with the published data and clarify the most important and novel points of this manuscript.

→ We thank the reviewer for this suggestion and have expanded the discussion on the most important and novel points of this manuscript. We expanded upon the novelty of our method, the novelty of the concerted conformational changes that appear as a conformational wave, and stress the achievement of observing TRPV1 opening by a single ligand. We also included the potential caveat/limitation of our studies in the discussion.

(3) In “Cryo-EM sample preparation and data collection” of the Methods part, the authors should describe more details about how to keep the samples at constant temperatures as this manuscript may set up a new method for further functional and structural study of temperature-sensitive TRPs. For example, how to ensure the samples can be kept in the specific temperature. Why are the samples incubated in a heat block just for 30 s but not incubated longer time?

→ Thank you for this suggestion. We have described more details regarding this point (updated the method section including the heat incubation time (lines 565-569).

Regarding the heat incubation time, we have tested longer incubation times, such as 45 sec, 50 sec, and 1 min 30 sec. However, we found that incubations over 45 sec resulted in severe protein degradation due to over-heating, therefore we suggest an incubation of just 30 s, which would likely require optimization for other systems.

REVIEWER COMMENTS

Reviewer #1 (Remarks to the Author):

Some of the concerns I raised on my previous review have been addressed, but the manuscript still contains serious conceptual inaccuracies that must be addressed, as well as lacking support from experimental densities for some of the conclusions. These concerns are described in detail below:

1) The structures obtained in the present manuscript reflect an equilibrium distribution of states at a given biochemical condition. A fundamental aspect of equilibrium distributions is that they do not offer any information about the timing of events, or the trajectory of conformational changes. All references to the temporality or trajectory of the observed conformational changes must therefore be removed, including the title, abstract and the main text.

Interpreting the observed conformational changes as a complete trajectory of concerted events neglects the dynamic nature of protein function. Certain conformational changes, such as the different rotamers in the methionine at position 644 that occlude the filter in some of the structures, could be highly dynamic and potentially occur in a non-concerted manner without a necessary correlation with some of the other conformational changes that are observed in other parts of the protein – the biochemical conditions (and cryogenic temperatures) might stabilize certain conformations that are otherwise dynamic when channels are in a biological membrane. Forcing an interpretation of a gating mechanism based on a few snapshots can have a negative impact in the field by simplifying a process that might be significantly more complex. This needs to be reflected in the interpretation of results throughout the manuscript.

2) Stating that the conformational trajectory observed in the manuscript is near complete is inaccurate, because as I pointed out in the first review, and also acknowledged in the response to reviewers, there is no certainty that certain conformations may have been missed in the observed ensembles. This is not a trivial detail. Moreover, it is a certainty that all intermediate conformations involving partially ligated states have been missed from the study. This needs to be addressed throughout the manuscript by toning down statements throughout. Adding a final disclaimer acknowledging these issues is in my opinion, not an adequate way to address this issue comprehensively.

3) It is stated in the manuscript that the temperature-dependence of RTx bound channels was measured experimentally, but this is inaccurate. As I pointed out in the original review, RTx-bound channels exhibit an open probability close to 1, and because of this, it is to be expected that temperature would not have an effect on P_o because it has already reached saturation. Macroscopic patch-clamp recordings such as those in Figure 1b reflect the activity of many ion channels that may be in different states, some of the

un-ligated and thus contributing to the high temperature dependence observed, and other ligated and lacking apparent temperature dependence because they are fully ligated. Therefore, the experiments don't really estimate the temperature dependence of RTx-bound channels. Because these experiments do not achieve what is expected of them, I think the authors should remove that data altogether from the manuscript. Importantly, data are shown for only a single experiment.

4) A structure including a density for M644 for the closed state should also be included in Figure 3c, as well as an analysis of the densities at different thresholding values to provide solid experimental support for the proposed conformational change.

5) Experimental densities should be included in Figure 3d.

6) Experimental densities must be included for the RTx ligand in Figure 5b. From looking at the extremely minor differences, it seems unlikely that there will be sufficient experimental support for the proposed configurational differences in the ligand.

7) Experimental densities must be included for the comparisons in Figure 5c.

8) It is unclear why different thresholding values were used for the densities in Figure 4. I think this could be potentially misleading. The same thresholding values should be used in all figures, or a more comprehensive analysis at different thresholding values should be provided to support the conclusions. This also applies for other figures where different thresholding values for the maps are used when performing comparisons between structures.

9) Experimental densities must be included in Figure 6a.

10) Experimental densities for the protein must be included in Supplementary Figure 7.

11) Experimental densities must be included in Supplementary Figure 9.

Minor points:

1) It is unclear where the data from Figure 1c comes from – I assumed those correspond to the experiment in Figure 1b, but this is unclear.

2) The reference to the phi-value analysis should be taken with care. This analysis is adequate for two-state systems, but not adequate for complex gating schemes where multiple states exist and where the states with major occupancy may differ as experimental conditions change. Importantly, in the cited reference not all phi-values observed are consistent with the interpretation provided, although those were not included in the main figures of the manuscript.

Reviewer #2 (Remarks to the Author):

The authors have addressed my concerns and answered most of my questions to my satisfaction, except for one -- question #7. The authors seemed confused the 3D auto-refinement with the generating the 3D reference map from the selected 2D classes. (3D initial model) . Therefore, the authors didn't answer my question: "Lines 482-483: Why the 3D auto refinement was performed before 3D classification?". In addition, the initial angular/translational assignments for each particle have been obtained through 2D classification, as author mentioned in the manuscript. Therefore, the authors still need to explain why the 3D auto-refinement was done before the 3D classification.

Reviewer #4 (Remarks to the Author):

No further comments

REVIEWER COMMENTS

Reviewer #1 (Remarks to the Author):

Some of the concerns I raised on my previous review have been addressed, but the manuscript still contains serious conceptual inaccuracies that must be addressed, as well as lacking support from experimental densities for some of the conclusions. These concerns are described in detail below:

1) The structures obtained in the present manuscript reflect an equilibrium distribution of states at a given biochemical condition. A fundamental aspect of equilibrium distributions is that they do not offer any information about the timing of events, or the trajectory of conformational changes. All references to the temporality or trajectory of the observed conformational changes must therefore be removed, including the title, abstract and the main text.

Interpreting the observed conformational changes as a complete trajectory of concerted events neglects the dynamic nature of protein function. Certain conformational changes, such as the different rotamers in the methionine at position 644 that occlude the filter in some of the structures, could be highly dynamic and potentially occur in a non-concerted manner without a necessary correlation with some of the other conformational changes that are observed in other parts of the protein – the biochemical conditions (and cryogenic temperatures) might stabilize certain conformations that are otherwise dynamic when channels are in a biological membrane. Forcing an interpretation of a gating mechanism based on a few snapshots can have a negative impact in the field by simplifying a process that might be significantly more complex. This needs to be reflected in the interpretation of results throughout the manuscript.

R) The reviewer requests that we remove claims that we have revealed a conformational trajectory of TRPV1 gating because we have performed our structural studies at equilibrium. We respectively point out that this assertion is incorrect. The reviewer must know the fact that equilibrium is temperature dependent. In our experiments, we poised the system at three equilibrium conditions using three different temperatures, so that relative state populations in each equilibrium condition should change. With heating, the equilibrium will naturally favor more toward the open state of this channel, owing to the positive Q_{10} . Therefore, conformational changes can readily be tracked using our novel method providing insight into the temporal sequence of conformations within the cryo-EM ensembles. As the reviewer points out, this does not provide kinetic information, but we make no such claims. The term “temporal” simply refers to the order of events as the channel opens. The reviewer’s claim that we forced the gating mechanism by using a few snapshots is not a fair assessment. Our work clearly shows many snapshots from the same cryo-EM ensemble and, importantly, using thermal titration we show that they are on the same conformational trajectory. This distinguishes our work from many conventional structural studies where multiple snapshots are captured from different biochemical conditions. Last, plunge freezing is well known to occur over sub microsecond timescales, so we are confident that we capture the motions of our proteins. The concerted rotameric movements of M644 are well addressed by our asymmetric cryo-EM reconstructions. Therefore, we have processed our structural and functional data with our model in a rigorous manner. Importantly, the other two reviewers are structural biologists and agree with our findings, calling our method novel and elegant. Without providing any evidence or scientific grounds against our structural data, but simply criticizing our data-driven claims as a negative impact in the field is not productive nor good science. We shall wait to see follow-up studies concerning our findings.

2) Stating that the conformational trajectory observed in the manuscript is near complete is

inaccurate, because as I pointed out in the first review, and also acknowledged in the response to reviewers, there is no certainty that certain conformations may have been missed in the observed ensembles. This is not a trivial detail. Moreover, it is a certainty that all intermediate conformations involving partially ligated states have been missed from the study. This needs to be addressed throughout the manuscript by toning down statements throughout. Adding a final disclaimer acknowledging these issues is in my opinion, not an adequate way to address this issue comprehensively.

R) We agree with the reviewer that we are undoubtedly missing conformations because of the continuum of conformations that proteins exist in. We do not claim to probe the complete dynamics of RTx-TRPV1, just the general trajectory for opening. Also, we want to once again emphasize that our studies start with the fully liganded TRPV1. Since the previous revision the reviewer has argued that the partially liganded TRPV1 is present. We responded that this is beyond the scope of our study. Having prepared TRPV1 in saturating conditions of RTx, it would be a rather rare event to have partial occupancy anyway, so such states would be rare indeed. We have addressed these points in our discussion, and also toned down the text by clarifying that we do not discuss the partially liganded channel.

The rewritten portion of the discussion follows (Lines 118-124):

“This is ideal for studying TRPV1 ligand-gating because under the saturating RTx conditions when RTx occupancy is maximized, thermal energy can be used to more gradually shift the conformation of TRPV1 toward the final open state without invoking the high noxious heat sensitivity. This may be because RTx binding ablates high noxious heat sensitivity or RTx-gating of TRPV1 dominates under this condition. In this condition we do not need to invoke partial occupancy and high heat sensitivity for our structural studies.”

3) It is stated in the manuscript that the temperature-dependence of RTx bound channels was measured experimentally, but this is inaccurate. As I pointed out in the original review, RTx-bound channels exhibit an open probability close to 1, and because of this, it is to be expected that temperature would not have an effect on P_o because it has already reached saturation. Macroscopic patch-clamp recordings such as those in Figure 1b reflect the activity of many ion channels that may be in different states, some of the un-ligated and thus contributing to the high temperature dependence observed, and other ligated and lacking apparent temperature dependence because they are fully ligated. Therefore, the experiments don't really estimate the temperature dependence of RTx-bound channels. Because these experiments do not achieve what is expected of them, I think the authors should remove that data altogether from the manuscript. Importantly, data are shown for only a single experiment.

R) The reviewer requests to remove the data in Fig.1 b because at a low RTx concentration, there are unliganded channels that exhibit high temperature sensitivity which complicates the interpretation. We want to re-emphasize that our studies are solely focused on TRPV1 with full RTx occupancy and its conformational trajectory using cryo-EM ensemble studies.

Importantly, the entire point of Fig. 1b is that when the RTX concentration is very high (fully bound), TRPV1 shows low temperature sensitivity $Q_{10} \sim 3$ (Fig. 1b). We then show that in our structural study RTx occupies all sites in all states of TRPV1, and therefore RTx-TRPV1 should follow this same temperature dependence of $Q_{10} \sim 3$ (Supplementary Figure 5). Whether Q_{10} is high at low RTx concentration because they are unliganded or liganded but show low temperature sensitivity is not

relevant and beyond our scientific focus in this study, which we have addressed in the previous revision. We only discuss fully bound RTX in our electrophysiology as well as cryo-EM studies. Finally, Fig. 1a and 1b are representative data for Fig 1c. Fig. 1c are two scatter plots with 34 data points from 17 cells. We have clarified this in the caption for Fig 1 (lines 824-826):

“c Q_{10} values as a function of $I/I_{50nM\ RTX}$ for low and high temperature ranges. Each experiment was conducted as shown in a and b. The low T range Q_{10} value is steady at 1.7, while the high T range Q_{10} rapidly collapses from ~ 38 to ~ 3 . Each pair of high and low temperature sensitivity data points represents independent time-course recordings from individual cells ($n = 17$ cells).”

4) A structure including a density for M644 for the closed state should also be included in Figure 3c, as well as an analysis of the densities at different thresholding values to provide solid experimental support for the proposed conformational change.

R) In our previous revision, we included a new figure 4 showing all the cryo-EM maps for key conformational changes in Fig. 3 including M644. To clarify further we have indicated key side chains in Fig. 4.

In cryo-EM, we provide the thresholding so that our map can be reproduced, but it is different from sigma level used in X-ray crystallography. We have adjusted all of our maps to use the same contour levels.

5) Experimental densities should be included in Figure 3d.

R) In our previous revision, we included a new Fig. 4 showing all the cryo-EM maps for key conformational changes in Fig. 3 including M644. To clarify further we have indicated key side chains in the revised Fig. 4.

The reviewer is now asking to provide cryo-EM density for every single figure. We have provided the cryo-EM density for the entire protein in the supplemental figures in addition to the Fig. 4 in our previous revision (see above). In addition, since the initial submission all the maps and coordinates were made available so that the reviewers could check the quality of our data. Showing cryo-EM density in every single figure is not always so helpful and in many cases makes figures cluttered and

confusing. However, we respect the reviewer's request, and have provided the cryo-EM density throughout the manuscript.

6) Experimental densities must be included for the RTx ligand in Figure 5b. From looking at the extremely minor differences, it seems unlikely that there will be sufficient experimental support for the proposed configurational differences in the ligand.

R) Please see the cryo-EM density maps that are provided in Fig. 5b. Again, we had provided all the maps and coordinates in our original submission for the reviewers' benefit. As can be seen in Fig. 5b, our RTx ligand densities are excellent without any ambiguity and with clear differences between binding poses. This region of the map has an average location resolution of 2.5 to 2.9 Å and the movement observed here is 0.8 to 1.9 Å, depending on the atom within RTx. In crystallography the coordinate precision at 2.5 to 2.9 Å is around 0.2 to 0.3 Å. Assuming crystallographic resolution and the cryoEM equivalent (based on our extensive experience in both areas) this level of precision holds in this case due to excellent map quality. So we think that based on any standard this movement is visually and objectively significant. Also, this really illustrates how subtle changes in ligand binding propagates and amplifies long-range protein conformation.

Fig. 5

7) Experimental densities must be included for the comparisons in Figure 5c.

R) Please see the cryo-EM density maps that are provided in Fig. 5c.

8) It is unclear why different thresholding values were used for the densities in Figure 4. I think this could be potentially misleading. The same thresholding values should be used in all figures, or a more comprehensive analysis at different thresholding values should be provided to support the conclusions. This also applies for other figures where different thresholding values for the maps are used when performing comparisons between structures.

R) In cryo-EM, we provide the thresholding so that our map can be reproduced, but it is different from sigma level used in X-ray crystallography. It can be difficult to precisely determine the sigma levels for Cryo-EM reconstruction maps, since multiple factors such as resolution, noise levels, map normalization, etc., could affect sigma values. We have adjusted all of our maps to the same contour levels to facilitate volume comparison, but they have different thresholding levels since the

reconstructions are from different datasets. We have clarified this point in the methods section by including the following statement:

“For figure generation, cryoEM density thresholds are adjusted to show the same contour level across maps.”

9) *Experimental densities must be included in Figure 6a.*

R) Please see the revised Fig. 6a.

10) *Experimental densities for the protein must be included in Supplementary Figure 7.*

R) Please see the revised Supp. Fig. 7.

11) Experimental densities must be included in Supplementary Figure 9.

R) Please see the revised Supp. Fig. 9.

48°C Heat-TRPV1 INT (7LPD)

RTx-TRPV1 IC

Minor points:

1) It is unclear where the data from Figure 1c comes from – I assumed those correspond to the experiment in Figure 1b, but this is unclear.

R) Thank you for bringing up this point. It corresponds to the Figure 1a (no RTx) and 1b (RTx). We have clarified this in the caption for Figure 1 by including the following statements:

For panel b: “**A representative** time-course recording for RTx-bound TRPV1 temperature sensitivity.” (Line 818)

For panel c: “ Q_{10} values as a function of $I/I_{50nM\ RTx}$ for low and high temperature ranges. **Each experiment was conducted as shown in a and b.** The low T range Q_{10} value is steady at 1.7, while the high T range Q_{10} rapidly collapses from ~ 38 to ~ 3 . **Each pair of high and low temperature sensitivity data points represents independent time-course recordings from individual cells ($n = 17$ cells).**” (Lines 824-827)

2) The reference to the phi-value analysis should be taken with care. This analysis is adequate for two-state systems, but not adequate for complex gating schemes where multiple states exist and where the states with major occupancy may differ as experimental conditions change. Importantly, in the cited reference not all phi-values observed are consistent with the interpretation provided, although those were not included in the main figures of the manuscript.

R) We thank the reviewer for pointing this out. We are aware of these points hence the reason we did not mention their studies in our original manuscript. We have thus toned down the referencing of this study in line 245 by simply saying "... consistent with the conformational wave idea proposed by previous studies.¹⁸"

Reviewer #2 (Remarks to the Author):

The authors have addressed my concerns and answered most of my questions to my satisfaction, except for one -- question #7. The authors seemed confused the 3D auto-refinement with the generating the 3D reference map from the selected 2D classes. (3D initial model) . Therefore, the authors didn't answer my question: "Lines 482-483: Why the 3D auto refinement was performed before 3D classification?". In addition, the initial angular/translational assignments for each particle have been obtained through 2D classification, as author mentioned in the manuscript. Therefore, the authors still need to explain why the 3D auto-refinement was done before the 3D classification.

R) We apologize for the confusion. To separate different conformations within the dataset, we explored multiple methods and utilized 3D classification without the image alignment technique. Since image alignment is not performed during this 3D classification, close-to-optimal angular/shift priors must be provided as the input. The best way to generate angular/shift priors is through a 3D refinement (we term consensus refinement). We found it more robust than SGD-based initial model generation methods (Relion's 3D initial model and CryoSPARC's *ab initio* reconstruction). Moreover, this 3D refinement step also helps us determine sample/data quality early since it provides a gold-standard resolution estimate. The 2D classification jobs do not generate 3D angular/shift priors, and we have made changes in the manuscript to clarify this. Furthermore, the output map from a consensus 3D refinement can be used as the 3D reference in the following 3D classification, rather than using a map from another dataset as to avoid potential reference bias.

We also added sentences to each data processing procedures section to explain the objective for the first 3D refinement in the main text. (lines 468, 510 and 551)

"... subjected to 3D auto-refinement to generate a 3D reference volume and find optimal orientations for subsequent 3D classification procedures..."

REVIEWERS' COMMENTS

Reviewer #1 (Remarks to the Author):

The authors have addressed most of my concerns, but there are still important issues that need to be resolved. My intent here is not to criticize the work itself, but to point out fundamental limitations that preclude drawing certain conclusions without appropriate experimental evidence. I will try to describe these more clearly:

1) The temperature-dependence of channels fully ligated by RTx is a very difficult quantity to measure. The Q10 of 3 measured in the presence of RTx in Figure 1 contains contributions from the temperature-dependence of ion diffusion ($Q_{10} \sim 1.5$) and of all channel species present in the ensemble (unliganded, partially ligated and fully ligated). The Q10 of RTx-ligated channels specifically remains a total unknown, but it most certainly is much smaller than 3. The data in Figure 1a-c therefore, provide no information about the temperature dependent distribution of states for fully ligated channels, but rather it simply shows that activated channels cannot be further activated.

Measurements of state occupancy at maximal channel stimulation have been obtained for voltage-gated potassium channels (e.g. <https://doi.org/10.1085/jgp.103.2.249>) and showed the existence of multiple off-pathway short-lived states. Identifying the number and connectivity between the open and the 'off-pathway' closed states required very extensive experimental characterization that included kinetic measurements. Multiple studies have addressed the question of how to identify the number of stably occupied states and their connectivity for a given mechanism (e.g. <https://doi.org/10.1085/jgp.84.4.505>), and all require kinetic measurements with high temporal resolution and lots of observed events. High-quality single channel recordings in TRPV1 fully bound to capsaicin ([https://doi.org/10.1016/S0006-3495\(03\)74719-5](https://doi.org/10.1016/S0006-3495(03)74719-5)) have provided evidence for a multiple open and closed states that can be occupied when channels are fully ligated, and multiple states have also been observed in the presence of RTx (<https://doi.org/10.1113/jphysiol.2005.087874>).

There is no information in the literature about the specific number of states that can become occupied in the case of TRPV1 channels when they are fully bound to RTx, and also there is no information about their temperature dependence when fully ligated. More importantly, no information is provided in this manuscript that would allow one to discriminate between the four models depicted below, and therefore no information is obtained about the trajectory or the timing of events.

a) $C \leftrightarrow IC \leftrightarrow O$

b) C \leftrightarrow O \leftrightarrow IC

c) IC \leftrightarrow C \leftrightarrow O

d) C \leftrightarrow O

\/
IC

That model (a) is the 'correct' one is an assumption made with no data to support it. This is pointed out correctly in some points of the manuscript: "We tentatively assume that class I, II, and III are closed, intermediate, and open states." (Page 7, lines 151-152).

However, there are other instances where the same statement is presented as a conclusion, which is incorrect and must be addressed in the text. Here are a few examples:

- Title: "The conformational trajectory of vanilloid-dependent TRPV1 opening revealed through cryoEM ensembles"

Should be rephrased as "A plausible conformational trajectory [...]" or something equivalent.

- Abstract: "we have temporally resolved the nearly complete conformational trajectory of the vanilloid receptor"

"RTx binding 42 to TRPV1 induces intracellular gate opening first, followed by selectivity filter dilation, then pore 43 loop rearrangement to reach the final open state."

- Results: "we concluded that the three conformational states in the cryo-EM ensembles of RTx-TRPV1 at 4°C represent distinct conformations along the RTx-dependent gating pathway and that their temporal sequence proceeds from class I through class II to class III, following their increasingly energetic states"

"Having resolved the order of transitions within the cryo-EM ensembles"

“Taken together, the conformational steps for RTx-dependent TRPV1 pore opening proceeds first through S6 gate opening, SF opening (via M644), then rearrangement of the PL and the outer pore (Fig. 3). The snapshots of these regions in the cryo-EM maps illustrate the conformational trajectory of RTx-dependent TRPV1 gating (Fig. 4).”

- Discussion: “We believe that our method of resolving the temporal sequence of conformations”

2) Given the large amount of evidence for the existence of multiple open and closed states of TRPV1 channels, it is arguably a certainty that three states do not represent the “near complete” conformational trajectory of activation of TRPV1 by any ligand.

Assuming that TRPV1 channel activation follows a general scheme as the one below:

C1 – C2 – C3 - ... Cn – O1 – O2 ... On

| | | | |

C2' – C3' ... Cn' – O1' – O2' ... On'

... ..

| | | | |

C2m – C3m ... Cnm – O1m – O2m... Onm

The dominant trajectories followed by individual channels may be different under varying conditions, and also different states would become occupied. The biochemical conditions have been found to largely impact channel behavior and to limit the number of states that can be observed. Some highly dynamic states, which could be major players under physiological conditions, could be poorly resolvable in cryo-EM grids, whereas very stable states that are rarely occupied in a cell membrane may become predominant in a cryo grid. If one followed a reasoning where the states observed in cryo-EM grids are truly representative of the occupancies of states under biological conditions, one would have to conclude that voltage-activated potassium channels never close because no closed structures of these proteins have ever been observed. Without any certainty or information about the conformations that are being missed, or their mechanistic relevance, it is formally impossible to draw any conclusions on whether a mechanism occurs in a concerted fashion or not. Therefore, claims about the completeness of the trajectory, or whether or not channels activate in a concerted fashion, should be removed from the manuscript.

Here are some examples:

Abstract: “This apparent conformational wave arises from the concerted, stepwise, additive structural changes of TRPV1 over many subdomains.”

Introduction: “our analysis has also revealed the near-complete conformational trajectory of RTX-dependent TRPV1 opening, including critical intermediate states, providing the basis for the long range allostery between RTX binding and pore opening.”

Results: “reveals that while localized (subdomain) conformational transitions occur in a concerted and stepwise manner, large long-range conformational changes appear to occur in a sequential manner in accordance with distance from the ligand-binding site.”

- Discussion: “we constructed the near complete energetic landscape for vanilloid-dependent TRPV1 gating”

Reviewer #2 (Remarks to the Author):

No further comments.

REVIEWERS' COMMENTS

Reviewer #1 (Remarks to the Author):

The authors have addressed most of my concerns, but there are still important issues that need to be resolved. My intent here is not to criticize the work itself, but to point out fundamental limitations that preclude drawing certain conclusions without appropriate experimental evidence. I will try to describe these more clearly:

1) The temperature-dependence of channels fully ligated by RTx is a very difficult quantity to measure. The Q10 of 3 measured in the presence of RTx in Figure 1 contains contributions from the temperature-dependence of ion diffusion (Q10 ~ 1.5) and of all channel species present in the ensemble (unliganded, partially ligated and fully ligated). The Q10 of RTx-ligated channels specifically remains a total unknown, but it most certainly is much smaller than 3. The data in Figure 1a-c therefore, provide no information about the temperature dependent distribution of states for fully ligated channels, but rather it simply shows that activated channels cannot be further activated.

Measurements of state occupancy at maximal channel stimulation have been obtained for voltage-gated potassium channels (e.g. <https://doi.org/10.1085/jgp.103.2.249>) and showed the existence of multiple off-pathway short-lived states. Identifying the number and connectivity between the open and the 'off-pathway' closed states required very extensive experimental characterization that included kinetic measurements. Multiple studies have addressed the question of how to identify the number of stably occupied states and their connectivity for a given mechanism (e.g. <https://doi.org/10.1085/jgp.84.4.505>), and all require kinetic measurements with high temporal resolution and lots of observed events. High-quality single channel recordings in TRPV1 fully bound to capsaicin ([https://doi.org/10.1016/S0006-3495\(03\)74719-5](https://doi.org/10.1016/S0006-3495(03)74719-5)) have provided evidence for a multiple open and closed states that can be occupied when channels are fully ligated, and multiple states have also been observed in the presence of RTx (<https://doi.org/10.1113/jphysiol.2005.087874>).

There is no information in the literature about the specific number of states that can become occupied in the case of TRPV1 channels when they are fully bound to RTx, and also there is no information about their temperature dependence when fully ligated. More importantly, no information is provided in this manuscript that would allow one to discriminate between the four models depicted below, and therefore no information is obtained about the trajectory or the timing of events.

a) $C \leftrightarrow IC \leftrightarrow O$

b) $C \leftrightarrow O \leftrightarrow IC$

c) $IC \leftrightarrow C \leftrightarrow O$

d) $C \leftrightarrow O$

\swarrow
 \searrow
IC

That model (a) is the 'correct' one is an assumption made with no data to support it. This is pointed out correctly in some points of the manuscript: "We tentatively assume that class I, II, and III are closed, intermediate, and open states." (Page 7, lines 151-152).

Response: The reviewer referred to past electrophysiological studies which suggest that ion channels sample multiple conformations including multiple open and closed states. Our structural studies are in line with these studies, by showing multiple closed and open states of fully RTX-bound TRPV1 in cryo-EM ensembles. Using thermal titration studies, we clearly show the temporal order of these conformational states that we resolved in our cryo-EM ensembles. Specifically, we derive the order of states as class I to class II to class III, which correspond to C, IC/IO, and O. Although our studies lack kinetic information and are not as high resolution as electrophysiological recordings, there is no assumption involved in our analysis. As the basis of our analysis we only rely on two facts, not assumptions: the temperature dependence of equilibrium and positive Q10 for TRPV1 gating).

Comment: *However, there are other instances where the same statement is presented as a conclusion, which is incorrect and must be addressed in the text. Here are a few examples:*

Response: We have toned down our statements as detailed below.

Comment: *Title: "The conformational trajectory of vanilloid-dependent TRPV1 opening revealed through cryoEM ensembles"
Should be rephrased as "A plausible conformational trajectory [...]" or something equivalent.*

Response: The title has been modified from "The conformational trajectory of vanilloid-dependent TRPV1 opening through cryoEM ensembles" to read "Vanilloid-dependent TRPV1 opening trajectory from cryoEM ensemble analysis."

Comment: *Abstract: "we have temporally resolved the nearly complete conformational trajectory of the vanilloid receptor"*

Response: We have modified the abstract from:

"By using thermal titration methods and cryo-EM we have temporally resolved the nearly complete conformational trajectory of the vanilloid receptor TRPV1 by the single modality action of the potent vanilloid resiniferatoxin (RTx)."

To:

"Here, we use thermal titration methods and cryo-EM in an attempt to obtain temporal resolution of the conformational trajectory of the vanilloid receptor TRPV1 with resiniferatoxin (RTx) bound." (Lines 42-44)

Comment: *"RTx binding to TRPV1 induces intracellular gate opening first, followed by selectivity filter dilation, then pore loop rearrangement to reach the final open state."*

Response: We have modified the text as such: "**We observe in our cryoEM ensemble analysis that** RTX binding to TRPV1 induces intracellular gate opening..." (Line 44)

Comment: Results: “we concluded that the three conformational states in the cryo-EM ensembles of RTx-TRPV1 at 4°C represent distinct conformations along the RTx-dependent gating pathway and that their temporal sequence proceeds from class I through class II to class III, following their increasingly energetic states”

Response: We modified the text as such: “... we **propose** that the three conformational states in the cryo-EM ensembles of RTx-TRPV1 at 4°C represent distinct conformations along the RTx-dependent gating pathway and that their temporal sequence proceeds from class I through class II to class III, following their **expectedly increasing** energetic states.” (Lines 159-162)

Comment: “Having resolved the order of transitions within the cryo-EM ensembles”

Response: We have modified the text from “Having resolved the order of transitions within the cryo-EM ensembles.” To: “Having **assigned** the order of **observed** transitions within the cryo-EM ensembles.” (Line 201)

Comment: “Taken together, the conformational steps for RTx-dependent TRPV1 pore opening proceeds first through S6 gate opening, SF opening (via M644), then rearrangement of the PL and the outer pore (Fig. 3). The snapshots of these regions in the cryo-EM maps illustrate the conformational trajectory of RTx-dependent TRPV1 gating (Fig. 4).”

Response: We have modified as such: “Taken together, **we propose that** the conformational steps for RTx-dependent TRPV1 pore opening proceeds first through S6 gate opening, SF opening (via M644), then rearrangement of the PL and the outer pore (Fig. 3). The snapshots of these regions in the cryo-EM maps illustrate **a plausible** conformational trajectory of RTx-dependent TRPV1 gating (Fig. 4).” (Lines 246-249)

Comment: Discussion: “We believe that our method of resolving the temporal sequence of conformations”

Response: The text was modified as such: “We believe that our **cryo-EM** method **for probing** the temporal sequence of conformations...” (Lines 369)

2) Given the large amount of evidence for the existence of multiple open and closed states of TRPV1 channels, it is arguably a certainty that three states do not represent the “near complete” conformational trajectory of activation of TRPV1 by any ligand.

Assuming that TRPV1 channel activation follows a general scheme as the one below:

C1 – C2 – C3 - ... Cn – O1 – O2 ... On
| | | | |

$C2' - C3' \dots Cn' - O1' - O2' \dots On'$

... ..

| | | | |

$C2m - C3m \dots Cnm - O1m - O2m \dots Onm$

The dominant trajectories followed by individual channels may be different under varying conditions, and also different states would become occupied. The biochemical conditions have been found to largely impact channel behavior and to limit the number of states that can be observed. Some highly dynamic states, which could be major players under physiological conditions, could be poorly resolvable in cryo-EM grids, whereas very stable states that are rarely occupied in a cell membrane may become predominant in a cryo grid. If one followed a reasoning where the states observed in cryo-EM grids are truly representative of the occupancies of states under biological conditions, one would have to conclude that voltage-activated potassium channels never close because no closed structures of these proteins have ever been observed. Without any certainty or information about the conformations that are being missed, or their mechanistic relevance, it is formally impossible to draw any conclusions on whether a mechanism occurs in a concerted fashion or not. Therefore, claims about the completeness of the trajectory, or whether or not channels activate in a concerted fashion, should be removed from the manuscript.

Response: It is a valid point the reviewer makes with regards to altered conformational landscapes and/or their sampling for channel in the membrane versus on a cryo-EM grid. Such effects are also dependent upon the system (the reference to the voltage-dependent ion channel is not appropriate because voltage gradient acts as a ligand, which cannot be applied in the *in vitro* cryo-EM system). As the reviewer mentioned, some systems in the purified form may have states inaccessible by conventional means. However, we have tried to ensure that our system behaves structurally as expected biochemically. For instance, we observe the channel in the closed and open states, with the various states and their equilibria following temperature dependence consistent with electrophysiology experiments. Further, our ability to resolve subtle conformational changes is exemplified by the ability to resolve differences in RTx binding poses as well as the selectivity filter conformations, including methionine flipping. While we could consider an overly complicated model with dozens of conformational states, we do not see such complexity in the data. Going with Occam's Razor, the simplest mechanism is often sufficient to explain the observations, and this is our approach. We do not discount the possibility of other states, but our data does not support it. As for our use of the word "concerted," we are referring to concerted motions of various subdomains at once, which (as far as we can resolve) is happening. We will certainly tone-down our claims of "complete" or "near complete" trajectories, but we will retain and clarify our use of "concerted," as detailed below.

Here are some examples:

Abstract: "This apparent conformational wave arises from 44 the concerted, stepwise, additive structural changes of TRPV1 over many subdomains."

Response: The text was modified as such: "This apparent conformational wave **likely** arises from the concerted, stepwise, additive structural changes of TRPV1 over many subdomains." (Line 47)

Introduction: "our analysis has also revealed the near-complete conformational trajectory of RTx-

dependent TRPV1 opening, including critical intermediate states, providing the basis for the long range allostery between RTx binding and pore opening.”

Response: The text was modified as such: “... our analysis has also revealed the ~~near-complete~~ conformational trajectory of RTx-dependent TRPV1 opening, including critical intermediate states, providing a basis for the long range allostery between RTx binding and pore opening.” (Lines 104-106)

Results: “reveals that while localized (subdomain) conformational transitions occur in a concerted and stepwise manner, large long-range conformational changes appear to occur in a sequential manner in accordance with distance from the ligand-binding site.”

Response: The text was modified from: “... reveals that while localized (subdomain) conformational transitions occur in a concerted and stepwise manner, large long-range conformational changes appear to occur in a sequential manner in accordance with distance from the ligand-binding site.” (Lines 264-266)

To: “... suggests that individual subdomain elements move together in a concerted manner, with large long-range conformational changes appearing to occur as sequential stepwise motions in accordance with the distance from the ligand-binding site.”

- Discussion: “we constructed the near complete energetic landscape for vanilloid-dependent TRPV1 gating”

Response: The text was modified as such: “... we constructed the ~~near-complete~~ energetic landscape for vanilloid-dependent TRPV1 gating.” (Lines 346)

Reviewer #2 (Remarks to the Author):

No further comments.